# Air Pollution and Cognitive Impairment across the Life Course in Humans: A Systematic Review with Specific Focus on Income Level of Study Area

**DOI:** 10.3390/ijerph19031405

**Published:** 2022-01-27

**Authors:** Mina Chandra, Chandra Bhushan Rai, Neelam Kumari, Vipindeep Kaur Sandhu, Kalpana Chandra, Murali Krishna, Sri Harsha Kota, Kuljeet Singh Anand, Anna Oudin

**Affiliations:** 1Department of Psychiatry, Centre of Excellence in Mental Health, Atal Bihari Vajpayee Institute of Medical Sciences (formerly PGIMER) and Dr. Ram Manohar Lohia Hospital, New Delhi 110001, India; chandra.b.rai@outlook.com (C.B.R.); kneelam1792@gmail.com (N.K.); sandhu.vipindeep@gmail.com (V.K.S.); 2Delhi Jal Board, Government of National Capital Territory of Delhi, New Delhi 110094, India; drkalpanachandra@gmail.com; 3JSS Academy of Higher Education & Research, Mysore 570015, Karnataka, India; muralidoc@gmail.com; 4Department of Civil Engineering, Indian Institute of Technology Delhi, New Delhi 110016, India; harshakota@civil.iitd.ac.in; 5Department of Neurology, Atal Bihari Vajpayee Institute of Medical Sciences (Formerly PGIMER) and Dr. Ram Manohar Lohia Hospital, New Delhi 110001, India; kuljeet_anand@rediffmail.com; 6Department of Public Health and Clinical Medicine, Umeå University, 901 87 Umea, Sweden; anna.oudin@med.lu.se; 7Department of Laboratory Medicine, Lund University, 901 87 Umea, Sweden

**Keywords:** air pollution, particulate matter_2.5_ (PM_2.5_), PAH, global pollution, health effects/risks, cognition, cognitive impairment, dementia

## Abstract

Cognitive function is a crucial determinant of human capital. The Lancet Commission (2020) has recognized air pollution as a risk factor for dementia. However, the scientific evidence on the impact of air pollution on cognitive outcomes across the life course and across different income settings, with varying levels of air pollution, needs further exploration. A systematic review was conducted, using Preferred Reporting Items for Systematic reviews and Meta-Analyses (PRISMA) Guidelines to assess the association between air pollution and cognitive outcomes across the life course with a plan to analyze findings as per the income status of the study population. The PubMed search included keywords related to cognition and to pollution (in their titles) to identify studies on human participants published in English until 10 July 2020. The search yielded 84 relevant studies that described associations between exposure to air pollutants and an increased risk of lower cognitive function among children and adolescents, cognitive impairment and decline among adults, and dementia among older adults with supportive evidence of neuroimaging and inflammatory biomarkers. No study from low- and middle-income countries (LMICs)was identified despite high levels of air pollutants and high rates of dementia. To conclude, air pollution may impair cognitive function across the life-course, but a paucity of studies from reLMICs is a major lacuna in research.

## 1. Introduction

Ambient air pollution is a leading cause of the global disease burden according to the Global Burden of Diseases, Injuries, and Risk Factors Study 2015, especially in low-income and middle-income countries [1]. Recent estimates ascribe 8.9 million deaths per year to ambient particulate matter (PM) having a diameter less than 2.5 microns (PM2.5) [2]. The PM2.5 from fossil fuels alone were, in another study, estimated to be a contributing cause to 10.2 million global excess deaths in 2012, with 62% of deaths in China (3.9 million) and India (2.5 million) [3]. Western Pacific and South-East Asia have the largest burden of disease related to air pollution worldwide, contributed by heavy industry and air pollution hotspots in the developing nations therein [4]. However, lower-middle income countries (LMICs) as well as low-income countries (LICs) have often been left behind when it comes to conducting epidemiological studies on air pollution health effects [5,6]. Unfortunately, it is not certain that results from epidemiological studies in high-income countries (HICs) or upper-middle-income countries (UMICs) can be directly extrapolated to LMICs or LICs, because both the quantities and sources of air pollution often differ between HICs/HMICs and LMICs/LICs, making the chemical composition of exposure, and subsequent health effects, unique. For example, in India (an LMIC), pollution is worse than in China (a UMIC). There are 22 Indian cities on the global list of the 30 most polluted cities. Apart from urban sources of air pollution, the burning of agricultural stubble in nearby rural areas also contributes to the burden of air pollution in Indian cities. In addition, in India all indicators of air pollution greatly exceed WHO standards [7], and concentrations are increasing [8]. Furthermore, African PM emissions often originate from old diesel-powered vehicles, and poor household waste management, and households burning biomass are the predominant contributors to outdoor air pollution [9]. In order to reduce uncertainties in the estimates for LICs and LMICs, epidemiological studies in these countries are, thus, needed [6].

Cognitive function, a prominent determinant of human capital, health, and socioeconomic status, is impacted by cumulative biological, social, and environmental exposures across the life course. Cognitive disorders such as dementia entail great suffering and high societal costs, and the prevalence worldwide is increasing. The number of people living with dementia is 55 million and is estimated to reach 75 million worldwide by 2030, with the majority living in LMICs and LICs. Recent studies have reported a decline in the prevalence of dementia in high-income countries, suggesting that dementia may, at least partially, be preventable [10,11].

Emerging studies suggest that exposure to air pollution may be associated with cognitive impairment, with reported effects ranging from impaired neurocognitive development in infancy and childhood to higher rates of cognitive decline and dementia in later life [12,13,14,15,16,17]. The Lancet Commission (2020) has recognized air pollution as a risk factor for dementia [18]. 

The aim and objective of this paper is to systematically review the evidence base with respect to the relationship between air pollution and cognitive health outcomes including dementia across the life course and in diverse income settings. There is a special focus on income level of the country of the study areas since LICs and LMICs often previously have been left behind when it comes to epidemiological studies of air pollution health effects. The high burden of cognitive disorders in LICs and LMICs, combined with the high burden of disease due to air pollution in these countries, highlights the need to make an inventory of epidemiological studies on air pollution in association with cognitive disorders in these countries. The contextualizing of research findings in terms of income settings of the research studies is valuable as the countries with lower income levels are disproportionately affected by air pollution while being resource-constrained to address either air pollution or its health impact. 

## 2. Materials and Methods

A systematic review was conducted to answer the research question on the impact of air pollution on cognitive health across the life course. 

The review took place between January and October 2020 based on the Preferred Reporting Items for Systematic reviews and Meta-Analyses (PRISMA) [19] Statement using a defined protocol that is unpublished (Appendix A). 

### 2.1. Search Strategy

A systematic search of the PubMed database was performed using PRISMA Guidelines with no time limit on the date of publication [19]. The Population, Investigated Exposure, Comparison, Outcome (PICO) Framework for this review is given in Table 1. 

The search string included keywords related to cognition and air pollution. Studies having any of the following keywords related to air pollution (exposure variable) and cognition (outcome variable) in their titles were identified. The keywords related to pollution utilized for PubMed search were Air Pollution, Pollutant (s), Particulate Matter, PM, Haze, Smog, Traffic-related air pollution (TRAP) and apportionment. The key words related to cognition utilized for PubMed search were Dementia, Cognitive, Memory, Attention, Cognition, Concentration, Orientation, Alertness, Alert, Intelligence, Emotion (s), Language, Reasoning, Planning, Decision making, Judgement, Recall, Learning, Emotion, Insight, Processing, Visuoconstructional, Coordination and Perception.

### 2.2. Selection Criteria

The inclusion criteria for the studies were full-text articles published in English with no time limit on date of publication, including original studies, systematic reviews and meta-analysis from any country on human participants of any age or gender. Protocols, letters to the editor and grey literature were excluded. 

Following the definition of the World Bank, country-level income was determined by a nation’s gross national income (GNI) per capita, where HICs, UMICs, LMICs and LICs had GNIs of USD > 12,375, 3996–12,375, 1026–3995, and < 1026 [20], respectively.

### 2.3. Screening Strategy

The search strategy comprised a two-stage process, In the first stage, three reviewers (C.B.R., N.K., V.K.S.) independently screened identified studies for eligibility by screening the titles and abstracts for study inclusion and exclusion criteria. References and citations of included papers were also reviewed to include additional potential articles. Abstracts of conference proceedings were searched for any relevant papers and posters. The second stage comprised independently screening the full texts of studies for eligibility by the three aforesaid reviewers. If the reviewers did not agree, then a more experienced reviewer was consulted (M.C.). 

### 2.4. Data Extraction and Analysis

Each included study was examined in detail. Data from the included studies were recorded for publication date, study setting, study type, duration of the study, study population, air pollution exposure and cognitive-outcome measures studied and reported. The results obtained were examined against the income level of the study area as per World Bank classification and contextualized. The extracted data were calibrated for consistency and completeness by independent reviewers (M.C. and K.C.). Due to heterogeneity in exposure and outcome variables and tools employed to assess these parameters, a statistical meta-analysis was not possible. Instead, the evidence was analyzed and presented based on nature of air pollutant, age profile, and gender of participants as well as income levels of study areas. 

### 2.5. Quality Assessment

The quality of individual studies was evaluated using the Revised tool for Quality Assessment on Diagnostic Accuracy Studies (QUADAS-2) [21]. The QUADAS-2 assesses the quality of primary diagnostic accuracy studies based on four key domains covering patient selection, index test, reference standard, and flow of patients through the study and timing of the index test(s) and reference standard (“flow and timing”). Each domain is evaluated for the risk of bias (using signaling questions), and the first three are also assessed for applicability. 

Two senior authors, not involved in the initial screening of studies for inclusion in the systematic review (vide supra), independently evaluated primary studies using QUADAS-2 (M.C., K.C.), and any difference in agreement was resolved by consensus with the third experienced author (M.K.), who was not involved in the initial screening of papers for eligibility criteria or quality evaluation. This ensured objectivity in the systematic review process.

Statistical methods for assessing publication bias using techniques such as funnel plots were considered but ruled out due to the heterogenous nature of available scientific literature on multiple exposures for multiple cognitive and neuroimaging outcomes across different age groups. We did not exclude any study based on quality. The results of analysis of publication bias are given in Table 2 and Figure 1. 

## 3. Results

### 3.1. Original Research Studies Obtained on Screening

A total of 1173 studies with a keyword related to cognition and a keyword related to pollution (in their titles) were obtained until 10 July 2020. The remaining 350 studies were manually reviewed for relevance (Rai, Sandhu, Kumari) to yield 53 original research studies (Figure 2).

### 3.2. Income Levels of Sites of Original Studies

The 53 original studies included 47 conducted in HICs and 6 conducted in LMICs. These are summarized in Table 3.

### 3.3. Research from LMICs/LICs

Among the studies, 89% were conducted in HICs, while only 11% of studies were conducted in UMICs. There was no original research from either LMICs or LICs as per World Bank income levels, despite high levels of air pollution (Appendix B).

### 3.4. Gender-Based Analysis

A review of 53 original studies revealed that most studies did not report gender-based findings (*n* = 28), and an equal number (*n* = 7) reported significant gender differences and no gender differences. Six studies were conducted on female participants, while five studies researched only male participants. (Figure 3).

### 3.5. Analysis Based on Life Course and Income Level of Study Areas

The results of the studies were analyzed based on the life-course stage (children and adolescents prenatal and postnatal exposures, adults and older adults) and on the income level of the study areas are presented below.

#### 3.5.1. Studies Related to Air Pollution and Cognitive Impairment in Children and Adolescents

##### Prenatal Exposure Studies

Prenatal exposure to different air pollutants has been analyzed for cognitive and other developmental indicators from infancy through adolescence using birth cohorts.

Multiple Air Pollutants

Two European studies found that exposure to multiple air pollutants in the prenatal period and infancy (until 1 year of age) had an adverse impact on psychomotor development and cognitive performance, adaptive functioning, and behavioral indices. 

Guxens et al. analyzed data from 9482 children from mother–infant pairs recruited between 1997 to 2008 in 6 European population-based birth cohorts, namely GENERATION R (Netherlands), DUISBURG (Germany), EDEN (France), GASPII (Italy), RHEA (Greece), and INMA (Spain) [54]. Prenatal exposures to NO_2_ and nitrogen oxides (NO_x_) for the exact pregnancy periods in all regions and PM (PM_2.5_, PM_10_, PM_2.5–10_, PM_2.5_ absorbance) in some regions were derived from background monitoring sites as well as land-use regression models for the period 2008 to 2011. Children were later assessed by a physician between 1 and 6 years of age. Prenatal air pollution exposure was associated with dose-dependent reduction in psychomotor development, where results of the different tests on average declined 0.68 points, with a 95% confidence interval (CI) of −1.25 to −0.11 per 10 μg/m^3^ increase in NO_2_. No such associations were seen for cognitive impairment. 

In another study from California, U.S., prenatal air pollutant exposure and first year of life exposure to NO_2_, PM_2.5_, PM_10_, ozone (O_3_), and near-roadway air pollution was associated with impaired cognitive performance, adaptive functioning, and behavioral indices among children with Autism Spectrum Disorder (ASD) (*n* = 327), but not severity of ASD [29]. However, third trimester PM_10_ exposure was paradoxically associated with improved behavioral performance in this study. 

A Mexican study also reported diffuse neuroinflammation, damage to the neurovascular unit and production of autoantibodies to neural and tight-junction proteins among children chronically exposed to higher concentrations of ozone and PM_2.5_ in Mexico. Such physiological responses enhance the risk of developing Alzheimer’s disease later in life [68].

2.Traffic-Related Air Pollution (TRAP)

Similarly, prenatal exposure to TRAP was found to have an adverse impact on intelligence quotient (IQ) in two separate birth cohort studies in the U.S. and Italy. 

Project Viva, a cohort of 1109 mother–child pairs in eastern Massachusetts, U.S.A, investigated the impact of prenatal and childhood exposure to TRAP, including black carbon (BC) and PM_2.5_ based on proximity to roadways and traffic density, on childhood cognition [47]. At a mean age of 8 years, children with a residence located less than 50 m away from a major highway had lower nonverbal IQ (−7.5 points; 95% CI: −13.1, −1.9), verbal IQ (−3.8 points; 95% CI: −8.2, 0.6) and visual–motor abilities (−5.3 points; 95% CI: −11.0, 0.4). Findings for cross-sectional associations between major roadway proximity and (1) prenatal and childhood exposure to traffic density, (2) prenatal and childhood exposure to PM_2.5_, and (3) third trimester and childhood exposure to BC were equivocal. In a study in Rome, Italy, the cognition of children aged 7 years (*n* = 474) who belonged to a birth cohort of 719 newborns was investigated [43]. The authors found that both traffic intensity within a 100 m buffer around the home and incremental increases (per 10 μg/m^3^) of intrauterine NO_2_ exposure were associated with reduced verbal IQ and verbal comprehension IQ (on Wechsler Intelligence Scale for Children-III). Other pollutants also showed negative associations but with much wider confidence intervals.

3.Particulate Matter (PM)

Prenatal PM exposure was found to have a deleterious impact on brain volumes in a neuroimaging study, demonstrating the neurological basis of impaired cognition in children exposed prenatally to air pollutants in previous studies.

A dose-dependent thinning of cerebral cortex per 5 μg/m^3^ increase in prenatal PM exposure in bilateral cerebral hemispheres was demonstrated in children living in Rotterdam, the Netherlands [14]. Reduced cerebral cortex in the precuneus and rostral middle frontal regions partially mediated the association between PM exposure and impaired inhibitory control. However, global brain volumes were not impacted.

4.Isophorone

Another cost-effective technique to assess exposure to multiple air pollutants is to study one indicator air pollutant, such as Isophorone (which is a common marker of industrial pollution), and its association with cognitive performance. Isophorone was investigated in a study on residential concentrations of 104 ambient air toxins from the National Air Toxics Assessment (2002). Machine learning algorithms were applied to the cognitive data of 6900 children from the Early Childhood Longitudinal Study Birth Cohort in the U.S.A. Here, a high Isophorone level (>0.49 ng/m^3^) was associated with low performance on mathematics tasks among children from urban and highly populated urban areas, with their scores being reduced by 1.19 points (95% CI: −1.94, −0.44) [37].

5.Persistent Organic Pollutants (POPs)

Another set of toxic air pollutants are POPs. Prenatal exposure to POPs such as polychlorinated biphenyls (PCBs), was investigated in adolescents (13–15 years) from 2 Dutch birth cohorts in the Development at Adolescence and Chemical Exposure (DACE) study [26]. 

Maternal serum levels of PCBs including 3 hydroxylated variants were measured during pregnancy [26]. There was a significant association between the compound PCB-183 and lower total intelligence, hexabromocyclododecane (HBCDD) and lower performance intelligence, and polybrominated diphenyl ethers (PBDEs) with lower verbal memory. Several hydroxy PCBs (OH-PCBs) were associated with more optimal sustained attention and balance. 

Among boys, poorer outcomes on fine motor skills and better outcomes on motor performance were reported, and prenatal DDE levels were associated with suboptimal motor performance, while a positive trend was seen between 4-OH-PCB-107 and fine motor skills. In girls, positive trends between OH-PCBs and verbal intelligence, and a negative trend between the compound BDE-153 and fine motor skills was observed. The associations persisted even after controlling for confounders such as maternal smoking, alcohol use, and education. 

6.Polyaromatic Hydrocarbons (PAH)

Prenatal exposure to PAH is another set of pollutants implicated in lower IQ, neuroimaging markers, impaired cognition in children and behavioral conditions such as Attention Deficit Hyperactivity Disorder (ADHD) and Conduct Disorder in separate birth cohort studies and different ethnic populations. 

A cohort involving prenatal personal ambient PAH monitoring was created with 665 urban Latina (Dominican) and African-American women, aged 18–35 years, residing in New York city [63]. The following inclusion criteria were applied: non-diabetic, non-hypertensive, not users of tobacco or illicit drugs, initial prenatal care by the 20th week of pregnancy, no known human immunodeficiency virus, low prenatal exposure to environmental tobacco smoke (cotinine levels < 15 µg/L in umbilical cord blood), and low exposure to pesticides (chlorpyrifos levels < 4.39 pg/g in umbilical cord blood). Results indicate that high prenatal PAH levels (>2.26 ng/m^3^) were associated with lower IQ (*p* = 0.007) and verbal IQ (*p* = 0.003), with offspring experiencing a decrement of 4.31–4.67 IQ points at 5 years of age. Neuroimaging of 40 randomly selected children in high and low prenatal PAH exposure groups demonstrated an inverse dose-response relationship with reduction of the white matter surface in the left hemisphere of the brain in later childhood (7–9 years), which is associated with slower information processing speed during intelligence testing and more severe externalizing behavioral conditions such as ADHD and Conduct Disorder [48]. 

Similarly, a study in Krakow, Poland, monitored healthy, nonsmoking pregnant women’s personal prenatal PAH levels (48 h) and examined blood samples and/or a cord blood samples at the time of delivery. After follow-up of the offspring (*n* = 214), the findings showed that high prenatal PAH exposure (>17.96 ng/m^3^) was associated with reduced nonverbal reasoning ability on Raven’s Coloured Progressive Matrices at 5 years of age, which corresponds to an average IQ decrease of 3.8 points [60].

##### Postnatal Exposure Studies

Most children continue to be exposed to air pollutants in postnatal periods as well. Several investigators have also explored the impact of postnatal exposure of the air pollutants known to have an adverse cognitive impact on prenatal exposure.

Multiple Air Pollutants

There has been original research on neuroimaging, neuropathological, and inflammatory markers as biomarkers of impaired cognition associated with postnatal exposure to multiple air pollutants.

Neuroimaging studies have also demonstrated the impact of high air-pollution exposure during the postnatal period on cognition. For instance, an MRI brain study found significant differences in white-matter volumes, especially in the right parietal and bilateral temporal areas, of children exposed to high levels of air pollution in Mexico City [69]. These same children had progressive deficits on the “Vocabulary” and “Digit Span” subtests of the Wechsler Intelligence Scale for Children-Revised (WISC-R).

Additionally, children with white-matter hyperintensities showed inflammatory markers, including evidence of inflammation resolution, immunoregulation, and tissue remodeling, while children without white-matter hyperintensities had raised levels of interleukin 12, pro-inflammatory cytokines, and low levels of neuroprotective cytokines and chemokines. The authors postulated that the air-pollution-induced neuroinflammation, characterized by complex modulation of cytokines and chemokines, causes structural and volumetric changes in the central nervous system (CNS) and, thereby, affects cognitive function. 

Another study on urban children supports air-pollution-associated neuroinflammation and neuropathology, as nearly 40% of highly exposed children and young adults had frontal tau hyperphosphorylation with pre-tangle material, and 51% had amyloid-beta diffuse plaques [68]. Conversely, 0% of the samples/scans from similar controls living in low-pollution areas showed such effects. 

2.Traffic-Related Air Pollution (TRAP)

Postnatal TRAP was found to be associated with impaired cognitive performance in four studies, though there is a contradictory study. All studies were from Spain.

Exposure to nitrogen dioxide (NO_2_), PM_2.5_, and elemental carbon (EC) derived from traffic over 1 year was studied in relation to the attention-related cognitive domain among 2687 primary school children from 39 schools across Barcelona, Spain [38,50]. Results demonstrated that children’s attention processes, assessed via a computerized, child-friendly Attention Network Test, were inversely associated with TRAP. Daily ambient levels of NO_2_ and PM_2.5_ contributed to an estimated equivalent of a 1.1-month (95% CI: 0.84, 1.37) delay in response speed according to the age-appropriate, natural developmental trajectory. Furthermore, boys demonstrated significantly worse cognitive performance on Hit Reaction Time following incremental increases of both EC and NO_2_.

Similarly, a study in Catalonia, Spain, explored the cognitive impact of TRAP exposure during the walking commute to school among children aged 7–10 years old (*n* = 1234) [12]. The authors found that exposure to PM_2.5_ and BC was associated with a reduction in the developmental growth of working memory; however, no significant association was seen for NO. 

A study conducted in Barcelona, Catalonia, Spain, also found an adverse impact of composite exposure to indoor and outdoor levels of TRAP at school, including EC, NO_2_, PM_2.5_ and ultrafine particles (UFP), on the development of working memory in children over 3.5 years of age (*n* = 2897) [34]. 

Conversely, Freire et al. found no significant association between NO_2_ and cognitive development in a population-based birth cohort of 210 children in Spain [61].

3.Persistent Organic Pollutants (POPs)

Similar to prenatal exposure, postnatal exposure to different POPs has been found to have an adverse cognitive impact. 

Lee et al. found a direct association between serum concentrations of POPs and the development of a learning disability (LD) and ADHD, both of which are characterized by cognitive impairment [65]. Children between the ages of 12 and 15 (*n* = 278) were recruited from the National Health and Nutrition Examination Survey in the U.S. (1999–2000). The most commonly detected POPs were 3,3’,4,4’,5-pentachlorobiphenyl, 1,2,3,4,6,7,8-heptachlorodibenzo-p-dioxin (HPCDD), 1,2,3,4,6,7,8,9-octachlorodibenzo-p-dioxin (OCDD), 1,2,3,4,6,7,8-heptachlorodibenzofuran (HPCDF), beta-hexachlorocyclohexane, p,p’-dichlorodiphenyltrichloroethane, and trans-nonachlor.

4.Isophorone

Like prenatal exposure, postnatal Isophorone has also been found to be associated with lower performance on mathematical tasks in children.

Ambient Isophorone exposure was found to be associated with low math scores (−1.48; 95% CI: −2.79, −0.18), independent of the influence of the home-learning environment, on analyzing the 2002 U.S. National Air Toxics Assessment data. Despite no interaction effect, children with both high Isophorone exposure and a low score on Home Observation for Measurement of the Environment Inventory had a decrement in math-scale score beyond the additive effect of individual exposures, especially in males [35]. 

#### 3.5.2. Studies Related to Air Pollution and Cognitive Impairment in Adults

Two studies from the U.S. and China reported impaired cognitive performance associated with exposure to air pollutants on cognitive performance in adults, which was contradicted by another study from the U.S. 

Chen and Schwartz found consistent associations in their study of 1764 adults in the United States aged 37.5 (±10.9 years) using the Third National Health and Nutrition Examination Survey (1988–1991) [62]. The findings demonstrated that long-term exposure to ambient ozone reduced performance on both the symbol–digit substitution test and a serial-digit learning test, but not in simple reaction-time tests. Each 10 parts per billion (ppb) increase in annual ozone was associated with the equivalent of 3.5 and 5.3 years of aging-related decline in cognitive performance (depending on the outcome measure used). Similar associations were not seen for PM_10_. Another study, in China, on the effects of cumulative and transitory exposure to air pollution on cognitive performance reported an adverse cognitive impact in verbal and math tests with greater deficits on verbal tasks as people aged [67].

However, results from the Reasons for Geographic and Racial Differences in Stroke (REGARDS) cohort found no consistent increase in the odds of cognitive impairment with every 10 µg/m^3^ increase in PM_2.5_ concentration (55). This was a geographically diverse, biracial U.S. cohort of both sexes (*n* = 20,150) aged ≥ 45 years with satellite-derived estimates of residential PM_2.5_ concentrations.

Further studies are required to clarify these contradictory findings, given that cognitive reserve in adults can mask minimal cognitive impairments. 

#### 3.5.3. Studies Related to Air Pollution and Cognitive Dysfunction in the Older Adults

Like childhood studies, there is extensive literature on the impact of air pollutants on cognition in the elderly.

##### Air and Noise Pollution

Several research studies have focussed on combined exposure of air and noise pollution, both of which are potentially deletrious for cognition in the elderly. 

A positive exposure-response relationship between dementia and PM_2.5_, NO_2_, but not ozone, was found in a cohort study in England (*n* = 30,978 aged 50–79 years) independent of noise levels [27]. Similarly, in the Betula project, a longitudinal study on health and ageing in Umeå, Sweden, the risk for incident dementia over a 22-year period between 1988 to 2010 was associated with higher residential exposure to NO_x_ (*n* = 1469 persons aged 60 to 85 years at inclusion) [42]. The authors reported a hazard ratio (HR) of 1.43 (95% CI: 0.998, 2.05) for the highest exposure quartile compared to the lowest [42]. The risk for dementia was independent of noise exposure, even in a relatively low-exposure area (*n* = 1721 persons aged 55–85 years) [25]. 

However, an additional study investigating associations within the entire Betula dataset (*n* = 1469 aged 60–85) showed no association between NO_x_ and episodic memory [36]. 

Research on a Heinz Nixdorf Recall population-based cohort (*n* = 4086 participants, aged 50–80 years from Bochum, Essen, and Mülheim in Germany) revealed that long-term residential exposure to size-fractioned PM, NO_x_, and traffic noise was negatively correlated with four neuropsychological subtests (verbal fluency, a labyrinth test, immediate verbal recall, and delayed verbal recall) as well as with global cognitive score (GCS) [44]. Here, the cognitive effects of dose-dependent increases in PM_2.5_ were independent of noise exposure. 

Tzivian et al., furthermore, found a positive dose-dependent association between long-term air pollution exposure and mild cognitive impairment (MCI), mainly amnestic subtype: odds ratio (OR) per interquartile range increase in PM_2.5_ of 1.16 (95% CI: 1.05, 1.27) [44]. In a subsequent study, this was found to be amplified by noise exposure, supporting the synergistic negative effect of air pollution and road traffic noise [40]. Similarly, exposure to traffic-related fine PM was also determined to be a consistent and significant risk factor for MCI in another German study [64]. 

##### Multiple Air Pollutants and TRAP

Similar to other age groups, multiple air-pollutant exposure has been determined to have an adverse impact on cognitive impairment and its concomitants in the elderly. 

Only one study has investigated the role of TRAP exposure in impairment of specific cognitive domains using a cohort-based approach. This study on older adults residing in Los Angeles reported an association between PM_2.5_ exposure and lower verbal learning (β = −0.32 per 10 μg/m^3^; 95% CI: −0.63, 0.00; *p* = 0.05). Moreover, NO_2_ exposure over 20 ppb was associated with lower logical memory, and ozone exposure above 49 ppb was associated with lower executive function [53]. Still, none of these pollutants were significantly associated with global cognition.

Similarly, a study on the association between TRAP and cognitive function in 680 elderly participants (aged 71 ± 7 years) in the U.S. Department of Veterans Affairs (VA) Normative Aging Study demonstrated a significant relationship between BC exposure and cognitive impairment, assessed using the Mini Mental Status Examination (MMSE) scores. The multivariable-adjusted OR was 1.3 for a doubling in BC concentration (95% CI: 1.1, 1.6) [59]. TRAP exposure was found to adversely impact reasoning and memory, but not verbal fluency, in a study on 2867 individuals (mean age: 66 years) from the Whitehall II cohort in the United Kingdom. Higher PM_2.5_ of 1.1 μg/m^3^ (lag 4) was associated with a 0.03 (95% CI: −0.06, 0.002) 5-year decline in standardized memory score [55]. In addition, in a study of 3377 elderly participants from the National Social Life, Health, and Aging Project (NSHAP) in the U.S., air pollutant exposure was estimated using GIS-based spatio–temporal models for PM_2.5_ and Environmental Protection Agency (EPA) monitors for NO_2_. Here, high PM_2.5_ exposure was associated with a 0.22 (95% CI: −0.44, −0.01) and high NO_2_ exposure with a 0.25 (95% CI: −0.43, −0.06) point decrease in Chicago Cognitive Function Measure test scores; these reductions are equivalent to aging 1.6 years and 1.9 years, respectively. The cognitive impact of PM_2.5_ was found to be mediated by depression and modified by stroke, anxiety, and stress. On the other hand, the cognitive impact of NO_2_ was mediated by stress, with impaired activities of daily living causing effect modification [39]. 

Another study by Molina-Sotomeyer et al. on 181 older women in Chile took into account the confounding effect of cardiorespiratory aerobic exercise for the impact of multiple air pollutants such as PM_10,_ PM_2.5_, NO_2,_ SO_2_ and Ozone concentrations on cognition and cardiovascular markers. The authors used maximum oxygen uptake (VO_2max_), estimated by the Six-Minute Walk Test (6mWT), heart rate (HR), and oxygen saturation (SpO_2_) as indicators of cardiorespiratory aerobic exercise. They marked significantly lower cognition performance amongst women residing in highly polluted areas as compared to those living in less-polluted areas. However, aerobic exercise was found to be a protective factor against adverse cognitive impact of air pollution, probably due to improvement in the mechanisms of oxygen transport [24].

Additionally, a large study on air pollution exposure and dementia incidence included all Canadian-born Ontario residents aged 55–85 years old and followed them from 2001 to 2013. The findings demonstrated that long-term mean residential exposure to air pollutants, including PM_2.5_, NO_2_ and ozone, was associated with higher dementia incidence, even at the relatively low exposure levels throughout this study setting. The HR was 1.04 (95% CI: 1.03, 1.05) for every interquartile-range increase in PM_2.5_ and 1.10 (95% CI: 1.08, 1.12) for every interquartile increase in NO_2_ [32]. Lo et al. also reported a significant association between long-term exposure to PM_10_ and O_3_ and cognitive impairment in older adults, with ORs of 1.094 (95% CI: 1.020, 1.174) and 1.878 (95% CI: 1.363, 2.560), respectively [23]. A greater effect was seen for the combined exposure to PM_10_ and O_3_ (*p* < 0.001). 

Further, research was conducted on the cross-sectional and cumulative (over 2.8 years) impact of outdoor air pollution exposure (PM and NO_x_) on cognitive functions of 86,759 middle- to older-aged adults (mean age 56.86 ± 8.12 years) from the U.K. Biobank general population cohort. While there was weak association between air pollution exposure and cognitive performance (dose-dependent slower reaction time, higher error rate on a visuospatial memory test, and lower numeric memory scores) at baseline (after adjusting for confounders), no such associations were found at follow-up. The authors did not report analyses stratified by sex [28].

##### Particulate Matter (PM)

PM has also been extensively investigated for a dose-response relationship with cognitive impairment in the elderly in several research projects in several high-income settings. 

Oudin et al. used a dataset of 1806 participants from the Betula project in Sweden (enrolled between 1993 and 1995 and followed until 2010) to assess the impact of exposures of ambient PM_2.5_ from residential wood burning and vehicle exhaust. Their results indicated a dose-dependent increased risk of incident dementia (Vascular Dementia and Alzheimer’s Disease) per 1 μg/m^3^ increase in PM_2.5_; reported HRs were 1.55 (95% CI: 1.00, 2.41; *p* = 0.05) for local wood burning and 1.66 (95% CI: 1.16, 2.39; *p* = 0.006) for traffic exhaust [30]. 

Similarly, PM_2.5_ exposure was associated with a 1.5 times greater cognitive error rate in a study on non-Hispanic, older White and Black U.S. adults (*n* = 780; age ≥ 55 years) from the 2001/2002 Americans’ Changing Lives Study. The adverse cognitive impact of PM_2.5_, was found to be similar among men and women, and possibly mediated by neighborhood social stressors and environmental hazards [31]. 

A neuroimaging study examining the association between PM_2.5_ and brain volumes in 1403 community-dwelling older women without dementia from the Women’s Health Initiative Memory Study (1996–1998) in Germany found that older women with greater PM_2.5_ exposure had significantly smaller white matter (WM) volumes in the frontal and temporal lobes and corpus callosum, which is equivalent to 1–2 years of brain ageing [45]. Similar associations were not seen for gray matter (GM) volumes or hippocampal volumes. 

Similarly, the 2004 Health and Retirement Study (*n* = 13,996; age ≥ 50 years) in U.S.A. also implicated high residential exposure to PM_2.5_ with worse cognitive function: β = −0.26 (95% CI: −0.47, −0.05) and strongest negative association for episodic memory [51].

Long-term exposure to high levels of PM_2.5–10_ and PM_2.5_ has been associated with faster dose-dependent cognitive decline in older women (aged 70 to 81 years) for every 10 μg/m^3^ increment in long-term (2 years) exposure in the U.S. Nurses’ Health Study Cognitive Cohort (*n* = 19409). The corresponding decrements on a global cognitive score following these PM_2.5–10_ and PM_2.5_ exposures were 0.020 (95% CI: −0.032, −0.008) and −0.018 (95% CI: −0.035, −0.002), respectively. The effect of 10 μg/m^3^ higher PM exposure was cognitively equivalent to aging by approximately 2 years [58]. 

Similarly, Saenz et al. also found that exposure to indoor air pollution, due to the household’s primary cooking fuel being wood or coal, was associated with poorer cognitive performance in older adults (over age 50) in Mexico [66]. 

Lee et al. reported that long-term exposure to a higher concentration of PM_2.5_ was associated with increased hospitalizations for dementia, with an adjusted HR of 1.049 (95% CI: 1.048, 1.051) per 1 μg/m^3^ increase in annual PM_2.5_. The hazard ratio for vascular dementia was somewhat higher at 1.086 (95% CI: 1.082, 1.090) [22]. Additionally, hospitalizations for dementia increased slightly as the level of urbanization increased: the HR for rural areas was 1.036 (95% CI: 1.031, 1.041) vis a vis 1.052 (95% CI: 1.050, 1.054) for the metropolitan areas.

The available literature strongly suggests that air pollution, especially PM, is associated with cognitive disorders in the elderly [70]. 

##### Polyaromatic Hydrocarbons (PAH)

There is only one study investigating the dose-dependent relationship between PAH and decline in cognitive performance. This study used the 2001–2002 National Health and Nutrition Examination Survey in the U.S. (*n* = 454) to demonstrate a dose-dependent association between PAH exposure per 1% increase (assessed by urinary 1-hydroxypyrene) and a decline in performance on digit symbol substitution tests by 1.8% among those aged ≥ 60 years [41].

##### Contribution of Genetic Factors

Cognitive impairment in the elderly may be associated with genetic factors such as polymorphisms in apolipoprotein E allele (ApoE4 variants) and hemochromatosis gene (*HFE C282Y* variant). Therefore, it is important that the confounding impact of such genetic factors are investigated to delineate the differential impact of air pollution and genetic loading, which has been investigated in some American and German studies.

Traffic load was a significant factor for cognitive performance among carriers of *ApoE ɛ4* risk alleles. Additionally, research using a cohort from the Women’s Health Initiative Memory Study (WHIMS) in the U.S. demonstrated that adverse cognitive effects of living in areas with high PM_2.5_ concentrations, including an 81% HR increase for global cognitive decline and a 92% HR increase for all-cause dementia, were exacerbated among ApoE *ɛ4/4* carriers [15]. Over a follow-up period of 22 years, Schikowski et al. examined the association between neurocognitive functions (assessed using CERAD-Plus) and air-pollution exposure as well as its modification by ApoE alleles among 789 women from the SALIA cohort in Germany [49]. Air pollution was inversely related to visuospatial abilities on figure copying tasks for every interquartile range increase of NO_2_ (β = −0.28 (95% CI: −0.44, −0.12)), NO_x_ (β = −0.25 (95% CI: −0.40, −0.09)), PM_10_ (β = −0.14 (95% CI: −0.26, −0.02)) and PM_2.5_ (β = −0.19 (95% CI: −0.36, −0.02)). 

In the Veterans Affairs Normative Aging Study from the U.S., Black Carbon (BC) was reported to be associated with lower cognition (*n* = 428 older men). Each doubling in BC levels was associated with 1.57 (95% CI: 1.20, 2.05) times higher odds of low MMSE scores in individuals with longer blood telomere length (OR = 3.23; 95% CI: 1.37, 7.59; *p* = 0.04 for BC-by-TL-interaction) [33], suggesting that genetic status may modify the effect of air pollution on cognitive health outcomes. The same study reported that older adults who lack an *HFE C282Y* variant (hemochromatosis gene polymorphisms) (*n* = 680; age 71 ± 7 years of age) are more susceptible to the adverse effects of TRAP exposure on cognitive function [58].

## 4. Discussion

Overall, the evidence from epidemiological studies points towards an association between exposure to pollutants and cognitive health effects across the life course. There is likely adverse cognitive impact of prenatal exposure to air pollutants, such as PAH, NO_2_, PM, POPs, with the evidence being strengthened by emerging neuroimaging research. Most studies also report an adverse cognitive impact of postnatal exposure to air pollutants in children, especially TRAP, POPs, and Isophorone. These associations have been validated by cognitive testing, neuroimaging studies and research on neuroinflammatory markers. 

The largest body of research is on the cognitive impact of air pollution among the elderly population. There is robust research demonstrating dose-dependent relationships between air pollutants such as PM, black carbon, TRAP, NO_2_, ozone, PAH and cognitive performance, neuroimaging markers and incident dementia. Furthermore, research on possible aetio-pathological mechanisms indicate that some genetic factors, including *APO E* and *HFE C28y* allele status, may intensify the adverse effects of air pollution on cognition, although evidence is not conclusive.

The research on associations between air pollution and cognitive health outcomes has occurred primarily in the HICs and UMICs of Europe and North America to Asia and Latin America, whereas there is paucity of such research in the LMICs and LICs of Africa and Asia, including India. No original research has been conducted on the cognitive impact of air pollutants in LMICs or LICs, even though the air pollution is generally much higher than the World Health Organization guidelines in LMICs/LICs as compared to HICs and UMICs [71]. Even the composition of particulate ambient air pollutants differs substantially, as their sources (household, industrial, traffic, etc.) may vary by setting. For example, indoor exposure is generally higher in LICs and LMICs than in UMICs and HICs, as woodstoves are often used for cooking. The extrapolations of the dose–response curves based on data from HICS and UMICs may not be applicable for LMICs and LICs. Therefore, generalizing results from HIC and UMICs to LICs and LMICs may be problematic due to varying levels, composition, and sources of air pollutants. The vast burden of both air pollution and neurocognitive disorders in LMICs, such as India with a 90% treatment gap [72], furthermore necessitates the synthesis of available evidence and systematic research to catalyze environmental policy change [73,74,75].

The proposed aetio-pathogenic pathways of air pollution’s effect on cognitive decline include enhanced risk of hypertension, dyslipidemia, oxidative stress, insulin resistance, endothelial dysfunction, procoagulant states, and stroke [76,77]. The plausible aetio-pathogenic pathways furthermore include structural changes in brain, neurodegeneration, and neuroinflammation. Increased levels of PM_2.5_ may be associated with smaller brain volumes, white matter lesions, higher rates of infarcts, and necrotic areas in the brain, all of which contribute to cognitive impairment [78]. Pathogenic mechanisms furthermore include hypoxia and direct neurotoxicity, which results from the breakdown of nasal, gut, lung epithelial, and blood-brain barriers, allowing an influx of airborne pollutants directly into the brain [79,80]. Secondary neurotoxicity can also occur when cytokines are transported from a lung injury site to the brain, resulting in neuroinflammation and neurodegeneration [79,80]. Air pollution, especially PM_2.5_, is also associated with the enhanced expression of pro-inflammatory mediators such as tumor necrosis factor (TNF-α) and interleukin-1β (IL-1β) as well as reactive oxygen species (ROS) [81]. The summary of proposed aetio-pathogenic impact of air pollution on cognition across the life course is given in Figure 4.

The quality of individual studies incorporated in this systematic review was evaluated using the Revised tool for Quality Assessment on Diagnostic Accuracy Studies (QUADAS-2) [21], and most research scored high on the quality of primary diagnostic accuracy. The risk of bias in the included studies on QUADAS-2 was low to moderate, considering that grey literature was not evaluated. The process of QUADAS-2 evaluation, by involving different researchers than the ones involved in primary screening, ensured objectivity. The studies scored low on risk of bias and applicability concerns for patient selection, index test, reference standard, flow of patients through the study, timing of the index test(s), and reference standard. This indicates that the diverse research on cognitive impact of air pollution across life course and across different income levels of sites of study is of good quality and can inform policy decisions.

Even though there is a growing evidence base for increasingly consistent results, dose–response relationships, and biological plausibility for air pollutants, particularly for exposure to PM_2.5_, the existing literature has certain limitations. The assessment of air pollution, although geocoded, may not reflect the true local exposure, as there is potential for variation even within a small geographical area. This can be further influenced by prevailing winds and seasonal patterns. Further, the individual exposure is only an estimate in these epidemiological studies and may not reflect the actual individual exposure that can be assessed in prospective studies using personal sensors. Moreover, it is a challenge to estimate and assess the effects of lifetime exposure to outdoor and indoor environmental pollutants in a mobile population, including occupational and transitory exposures, with complete data on exposure concentration, duration, pollutant type, and source apportionment. As the included studies utilized a variety of exposure assessment methods, such as dispersion models and land-use regression modelling, the data were too disparate to combine into a meta-analysis. Unlike cohort based epidemiological studies, case-finding bias may occur in hospital-based studies as participants may already have health concerns associated with exposure to air pollution (for example Bronchial Asthma and Chronic Obstructive Pulmonary Disease) necessitating their visit to the hospital. Adequate assessment of confounding factors, such as genetic factors (such as ApoE ɛ*4*), cardiometabolic factors, individual and parental socioeconomic and educational status, and living conditions, is also required. Finally, there may be an emerging publication bias for studies reporting significant associations as this area expands. 

This review has synthesized existing scientific evidence regarding the association between exposure to air pollution and cognitive health outcomes across the life course. However, the causal pathway of this relationship is still not fully understood, although evidence on aetio-pathogenic pathways is rapidly increasing from experimental studies. Establishing the causal effects of air pollution on cognitive health outcomes will likely require international, multicentric cohort-based research with harmonized protocols that periodically assess air pollutant exposures, cognitive function, systemic cardio–metabolic illnesses, neuroimaging, and inflammatory and genetic markers. In their 2015 systematic review of 13 studies, Peters et al. advocated for continued research on the association between air pollution and cognitive decline among older adults [81]. Additionally, two Lancet Commissions have supported emerging evidence for a causal role of PM in impaired cognition in midlife and later life, and have recommended further research [18,75]. As air pollution is a preventable risk factor, its regulation has potential health and fiscal benefits for individuals and society. Further scientific evidence (including life-course research, especially in LICs and LMICs) is required to bring about policy change for improved air quality and, thereby, protect cognition across the life course. 

## 5. Conclusions

This systematic review of all published original studies on the cognitive impact of air pollution across the life course (using PRISMA guidelines) demonstrates that different air pollutants have an adverse cognitive impact across the life course. Most research is confined to HICs and UMICs with no original research in LMICs and LICS despite high rates of both air pollutants and cognitive impairment/dementia in these settings. Hence, the extrapolations of the dose–response curve based on data from HICS and UMICs may not be applicable for LMICs and LICs. As air pollution is a preventable risk factor, its regulation has potential health benefits and resultant cost-saving by potentially improving cognitive health and reducing the risk of dementia later in life. Further life-course research, especially in LICs and LMICs, is needed to establish aetio-pathogenic pathways for differential levels and cumulative exposure to air pollution, to promote policy change for air pollution and its health effects, including adverse cognitive impact. 

## Figures and Tables

**Figure 1 ijerph-19-01405-f001:**
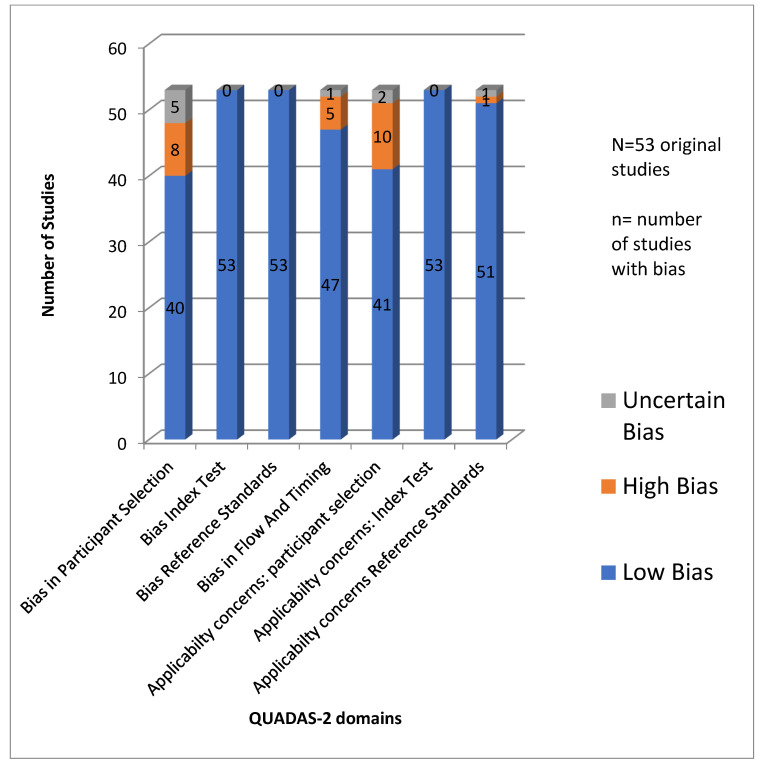
Analysis of publication bias metrics for literature on cognitive impact of air pollution using QUADAS-2.

**Figure 2 ijerph-19-01405-f002:**
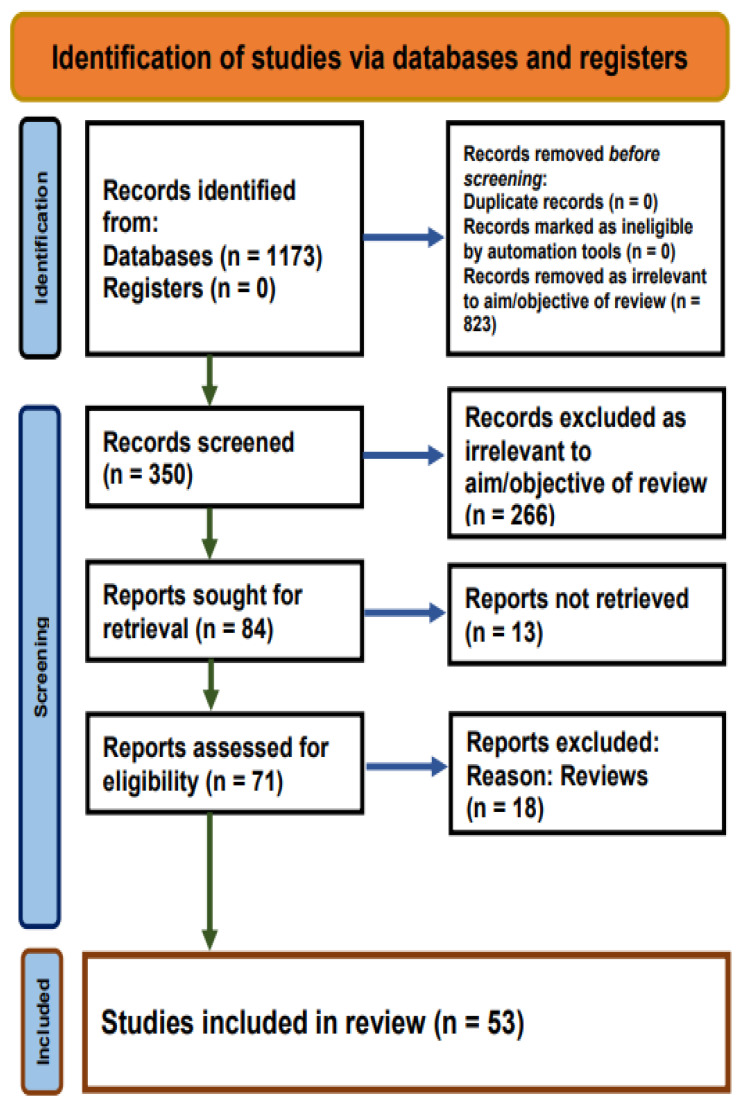
PubMed search results for studies on Air Pollution and Cognition across Life-Course.

**Figure 3 ijerph-19-01405-f003:**
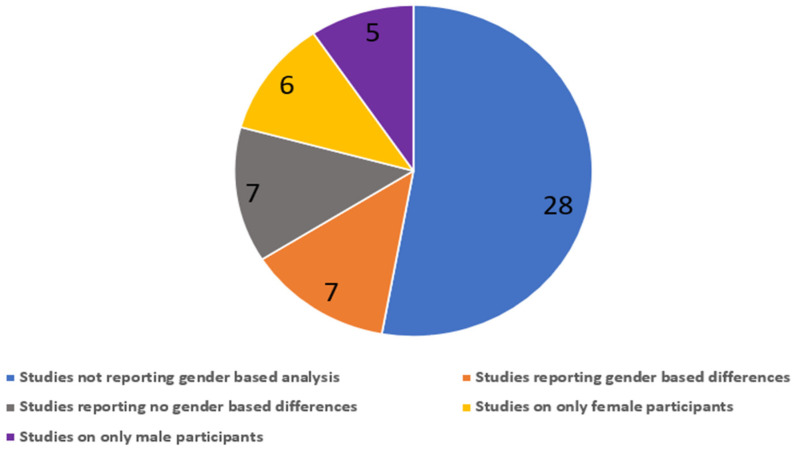
Gender-based reporting in original studies on air-pollution-associated cognitive impairment.

**Figure 4 ijerph-19-01405-f004:**
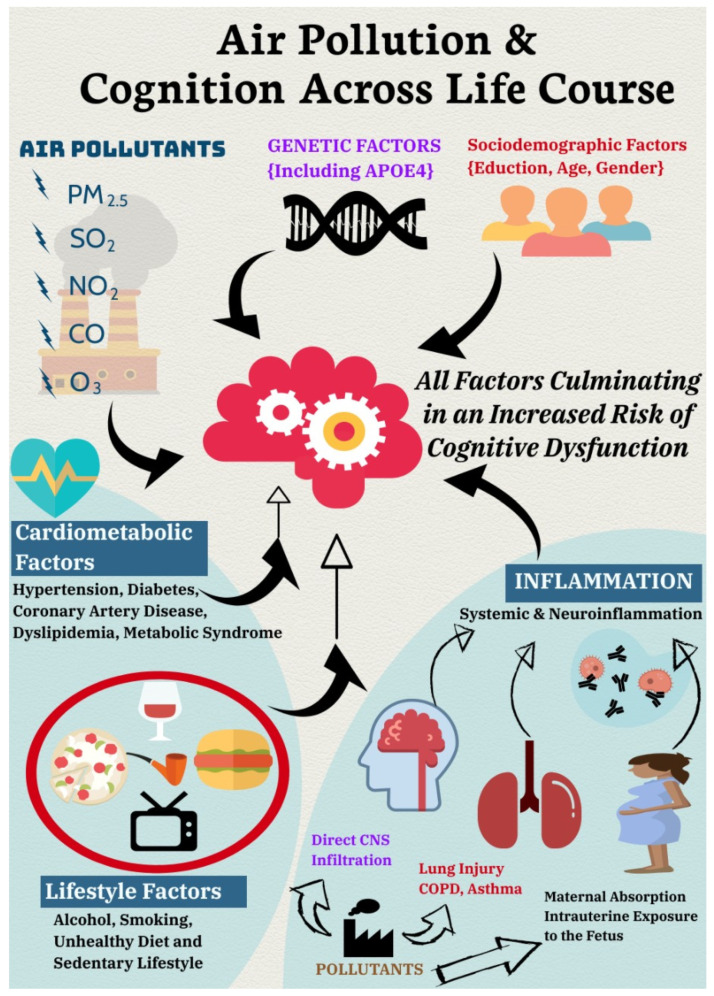
Proposed aetio-pathological mechanisms of cognitive impact from air pollution across the life course (made using piktochart.com).

**Table 1 ijerph-19-01405-t001:** The Population, Investigated Exposure, Comparison, Outcome (PICO) Framework for Research Question.

Population	Any Age Group Both Genders Participant Level Data or Cohort Level Data
Investigated exposure	Air Pollution: single or multiple Cross sectional or Cumulative exposure In utero exposure, Circumscribed cross sectional exposure or Lifetime exposure
Comparison	Air Pollutant exposed population comparators Gender comparators Income settings comparators
Outcome	Cognition and its different domains, neuroimaging markers

**Table 2 ijerph-19-01405-t002:** Analysis of publication bias in published literature on cognitive impact of air pollution using the QUADAS-2 framework.

S. No.	Study: Author (Year)	Risk of Bias Low: 1, High: 2, Uncertain: 3				Applicability Concerns Low: 1, High: 2, Uncertain: 3		
		Participant Selection	Index Test	Reference Standards	Flow and Timing	Participant Selection	Index Test	Reference Standards
A	Studies from High Income Settings							
1	Lee et al. (2019) [22]	1	1	1	1	1	1	1
2	Lo et al. (2019) [23]	1	1	1	1	1	1	1
3	Molina-Sotomayor et al. (2019) [24]	2	1	1	2	1	1	1
4	Andersson et al. (2018) [25]	1	1	1	1	2	1	1
5	Berghuis et al. (2018) [26]	2	1	1	1	2	1	1
6	Carey et al. (2018) [27]	1	1	1	1	1	1	1
7	Cullen et al. (2018) [28]	1	1	1	1	1	1	1
8	Guxens et al. (2018) [14]	1	1	1	1	1	1	1
9	Kerin et al. (2018) [29]	2	1	1	1	2	1	1
10	Oudin et al. (2018) [30]	1	1	1	1	1	1	1
11	Ailshire et al. (2017) [31]	3	1	1	1	2	1	1
12	Alvarez-Pedrerol (2017) [12]	1	1	1	1	1	1	1
13	Cacciottolo et al. (2017) [15]	1	1	1	1	1	1	1
14	Chen et al. (2017) [32]	1	1	1	1	1	1	1
15	Colicino et al. (2017) [33]	2	1	1	1	2	1	1
16	Forns et al. (2017) [34]	1	1	1	1	1	1	1
17	Lett et al. (2017) [35]	1	1	1	1	1	1	1
18	Oudin et al. (2017) [36]	1	1	1	1	1	1	1
19	Stingone et al. (2017) [37]	1	1	1	1	1	1	1
20	Sunyer et al. (2017) [38]	1	1	1	1	1	1	1
21	Tallon et al. (2017) [39]	1	1	1	1	1	1	1
22	Tzivian et al. (2017) [40]	1	1	1	1	1	1	1
23	Best et al. (2016) [41]	1	1	1	1	1	1	1
24	Oudin et al. (2016) [42]	1	1	1	1	1	1	1
25	Porta et al. (2016) [43]	3	1	1	1	3	1	1
26	Tzivian et al. (2016) [44]	1	1	1	1	1	1	1
27	Tzivian et al. (2016) [45]	1	1	1	1	1	1	1
28	Chen et al. (2015) [46]	1	1	1	1	1	1	1
29	Harris et al. (2015) [46]	1	1	1	1	1	1	1
30	Peterson et al. (2015) [47]	1	1	1	1	1	1	1
31	Schikowski et al. (2015) [48]	1	1	1	1	1	1	1
32	Sunyer et al. (2015) [49]	1	1	1	1	1	1	1
33	Ailshire and Clarke (2014) [50]	3	1	1	3	2	1	1
34	Ailshire and Crimmins (2014) [51]	1	1	1	1	1	1	1
35	Gatto et al. (2014) [52]	1	1	1	1	1	1	1
36	Guxens et al. (2014) [53]	1	1	1	1	1	1	1
37	Tonne et al. (2014) [54]	1	1	1	1	1	1	1
38	Loop et al. (2013) [55]	1	1	1	1	1	1	1
39	Power et al. (2013) [56]	1	1	1	1	1	1	1
40	Weuve et al. (2012) [57]	1	1	1	1	1	1	1
41	Power et al. (2011) [58]	1	1	1	1	1	1	1
42	Edwards et al. (2010) [59]	1	1	1	1	1	1	1
43	Freire et al. (2010) [60]	3	1	1	1	2	1	1
44	Chen and Schwartz (2009) [61]	1	1	1	1	1	1	1
45	Perera et al. (2009) [62]	1	1	1	1	1	1	1
46	Ranft et al. (2009) [63]	3	1	1	1	1	1	1
47	Lee et al. (2007) [64]	1	1	1	1	1	1	1
B	Studies from Upper Middle Income Settings							
1	Saenz et al. (2018) [65]	1	1	1	1	1	1	1
2	Zhang et al. (2018) [66]	1	1	1	1	1	1	1
3	Calderón-Garcidueñas et al. (2016) [67]	2	1	1	2	3	1	3
4	Calderón-Garcidueñas et al. (2012) [68]	2	1	1	2	2	1	1
5	Calderón-Garcidueñas et al. (2011) [69]	2	1	1	2	2	1	2
6	Calderón-Garcidueñas et al. (2008) [13]	2	1	1	2	2	1	1

**Table 3 ijerph-19-01405-t003:** Summary of original research on cognitive impact of air pollution across life course.

S. No.	Author (Year)	Study Site	Study Population	Exposure Studied	Outcome Variable Studied	Cognitive Impact
A	Studies from High Income Settings					
1	Lee et al. (2019) [22]	South-eastern part of the United States	94 million follow-up records from fee-for-service Medicare records for 13 million Medicare beneficiaries of fee for service (FFS) residing in the southeastern United States (U.S.) from 2000 to 2013.	Spatially and temporally continuous PM_2.5_ exposure data	Hospitalization rates for dementia	Long-term exposure to a high PM_2.5_ levels associated with increased hospitalization with dementia per 1 μg/m^3^ increase in annual PM_2.5_, with higher risk for vascular dementia. Did not report analysis stratified by sex
2	Lo et al. (2019) [23]	Taiwan	2241 community-dwelling, free-living elderly population with mean age at the time of recruitment 73.62 years; M:F = 57.5:42.5) followed from 1996 to 2007	PM_10_ Ozone	Short Portable Mental Status Questionnaire	Long-term exposure to PM_10_ and O_3_ associated with cognitive impairment with greater impact of the joint effect of exposure to PM_10_ and O_3_ Did not report analysis stratified by sex
3	Molina-Sotomayor et al. (2019) [24]	Chile	181 older women, patients of the “La Estrella” Health Center of the Pudahuel commune, Metropolitan Region of Santiago de Chile, and patients of the Senior Centers of Viña del Mar City-Chile	Average PM_10_ PM_2.5_ NO_2_ SO_2_ Ozone Concentrati-on between 2012–2014 Cardio-respiratory Aerobic exercise indicators: Maximum oxygen uptake (VO_2max_), estimated by the Six-Minute Walk Test (6mWT); heart rate (HR); and oxygen saturation (SpO_2_).	Mini Mental State Examination (MMSE)	Significant differences (*p* < 0.05) between the Active Group residing in relatively less polluted areas versus sedentary group residing in more polluted areas on all the MMSE dimensions except “Registration”, and in all the physiological variables (VO_2max_, SpO_2_, HR). Aerobic exercise may be a protective factor adverse cognitive effects of air pollution have on cognition probably due to improvement in the mechanisms of oxygen transport.
4	Andersson et al. (2018) [25]	Umea, Sweden	Data of 1721 participants aged 55–85 years at baseline (Male: Female = 985:736 of Betula project, a longitudinal study of health and ageing aged 55–85 years at baseline	Estimates of annual mean levels of nitrogen oxides (NO_x_) at the participants’ residential address using a land-use regression model. Modelled data for road traffic noise levels at the participants’ residential address	Dementia incidence	302 of 1721 participants at baseline, 302 developed dementia during the follow up period. Residing in the two highest quartiles of NO_x_ exposure was associated with an increased risk of dementia which was not modified by adjusting for noise. Did not report analysis stratified by sex
5	Berghuis et al. (2018) [26]	Nether-lands	101 children aged 13–15 years (M:F 55:46) participating in Development at Adolescence and Chemical Exposure (DACE)-study a follow-up of two Dutch birth cohorts.	Maternal pregnancy serum levels of PCB-153 and three OH-PCBs, 9 PCBs, 5 polybrominated diphenyl ethers (PBDEs), dichloroethene (DDE), pentachlorophenol (PCP) and hexabroomcyclododecane (HBCDD) in different parts of the cohort Maternal smoking and alcohol use	Wechsler Intelligence Scale for Children, third edition, Dutch version (WISC-III-NL) Dutch version of the Rey’s Auditory Verbal Learning Test (AVLT) Movement Assessment Battery for Children (Movement-ABC)	Significant association between PCB-183 and lower total intelligence, HBCDD with lower performance intelligence and PBDEs with lower verbal memory Positive trends between OH-PCBs and verbal intelligence, and a negative trend between BDE-153 and fine motor skills was observed Several OH-PCBs associated with more optimal sustained attention and balance CONTD ON NEXT PAGE Boys had poorer outcomes on fine motor skills and better outcomes on ball skills with positive trend was seen between 4-OH-PCB-107 and fine motor skills In girls, Prenatal DDE levels were associated with (sub)clinical motor performance
6	Carey et al. (2018) [27]	London, UK	Retrospective cohort of 130,978 adults aged 50–79 years	Average annual concentrations of NO_2_, PM_2.5_, ozone (O_3_), traffic intensity, distance from major road and night-time noise levels	Clinical diagnosis of Dementia	Positive exposure-response relationship between dementia and all measures of air pollution except O_3_. Did not report gender based findings
7	Cullen et al. (2018) [28]	UK	86,759 middle- to older-aged adults from the UK Biobank	Cumulative impact of outdoor air pollution exposure (PM_10_, PM_2.5_, NO_2_, NO_x_)	Cognitive function including reasoning test, pairs matching test, reaction time, prospective memory, visuospatial memory, numeric memory	Weak association between air pollutant exposure and cognitive performance at baseline (dose-dependent lower reaction time, higher error rate on a visuospatial memory test and lower numeric memory scores) with no such association at 2.8 years follow-up. Did not report analysis stratified by sex
8	Guxens et al. (2018) [14]	Nether-lands	Data from population-based birth cohorts—GENERATION R (The Netherlands) (2002–2006) that recruited mother infant dyads including 8879 pregnant women and 1932 children born between April 2002 and January 2006 were taken of which 783 children between ages of 6–10 years participated in MRI sub-study	Prenatal exposure to air pollutants such as (NO_2_, NO_x_) in all regions and PM (PM_2.5_, PM_10_ and PM _coarse_) and PM_2.5_ absorbance in a subgroup using land-use regression models for the period 2008 to 2011 and then modelled for air pollutant profile for the exact pregnancy periods using background monitoring sites.	MRI Brain Cognitive and psycho-motor development	Air pollution exposure was not associated with global brain volumes Higher prenatal PM exposure per 5 μg/m^3^ had a dose-dependent thinning of cortex increase in several brain regions of both hemispheres (e.g., cerebral cortex of the precuneus in Right hemisphere). Reduced cerebral cortex in precuneus and rostral middle frontal regions partially mediated the association between PM exposure and impaired inhibitory control. Did not report analysis stratified by sex
9	Kerin et al. (2018) [29]	California, USA	327 children with Autism Spectrum Disorder (ASD) from the Childhood Autism Risks from Genetics and the Environment study	Exposure to NO_2_, PM_2.5_ and PM_10_, ozone, and near-roadway air pollution in each trimester of pregnancy and first year of life.	Mullen Scales of Early Learning (MSEL), the Vineland Adaptive Behavior Scales (VABS), and the Autism Diagnostic Observation Schedule calibrated severity score.	ASD severity not associated with any air pollutant exposure. Prenatal and First year exposure to NO_2_ associated with impaired cognitive performance, adaptive functioning and behavioral indices but not severity of ASD 3rd Trimester PM_10_ exposure was paradoxically associated with improved behavioral performance Did not report analysis stratified by sex
10	Oudin et al. (2018) [30]	Umea, Norhern Sweden	1806 participants from Betula project from Umeå, Northern Sweden enrolled between (1993–1995) and followed upto 2010	Modelled levels of source-specific residential fine PM exposure to wood stoves or wood boilers and traffic	Validated data on dementia diagnosis	Increased dose-dependent risk for incident dementia (Vascular Dementia and Alzheimer’s Disease) with local residential wood burning and traffic exhaust Did not report analysis stratified by sex
11	Ailshire et al. (2017) [31]	USA	779 U.S. adults age ≥ 55 years from the 2001/2002 wave of the Americans’ Changing Lives study	Annual average PM_2.5_ concentration in 2001 in the area of residence by linking respondents with EPA air monitoring data using census tract identifiers. Exposure to neighborhood social stressors using perceptions of disorder and decay including subjective evaluations of neighborhood upkeep, presence of deteriorating/abandoned buildings, trash, and empty lots.	Error rate on Short Portable Mental Status Questionnaire (SPMSQ).	Association between higher rates of cognitive errors with high concentrations of PM_2.5_ which was stronger in high stress neighborhoods indicating towards a possible role of social stressors and environmental hazards No statistically significant gender based differences found
12	Alvarez-Pedrerol (2017) [12]	Barcelona, Spain	1234 children aged 7–10 years from 39 schools who commuted to school by foot	TRAP exposure (Average PM_2.5_, Black Carbon (BC) and NO_2_ concentrations) for the shortest walking route to school	Working Memory (the three-back numbers test) and inattentiveness (hit reaction time standard error of the Attention Network Test)	PM_2.5_ and Black Carbon were associated with a reduction in the growth of working memory with no significant association of working memory with NO_2_ Associations of TRAP concentrations (BC and PM_2.5_) and working memory were stronger for males than females
13	Cacciottolo et al. (2017) [15]	USA	3647 women aged 65 to 79 years from Women’s Health Initiative Memory Study (WHIMS)	PM_2.5_	Global cognitive decline and all-cause dementia APOE ɛ4/4 status	Residence in places with high PM_2.5_ was associated with an increased risk for global cognitive decline and all-cause dementia by 81% and 92% respectively, with risk exacerbated by APOE ɛ4/4 allele status
14	Chen et al. (2017) [32]	Ontario, Canada	All Ontario residents who were 55–85 years old on 1 April 2001, Canadian-born, and free of physician-diagnosed dementia who were followed up to 2013	Long-term average residential exposure to Air Pollutants including PM_2.5_, NO_s_ and ozone	Dementia incidence	Every interquartile-range increase in exposure to M_2.5_ and NO_2_ even at low levels of air pollution associated with higher incidence of dementia No statistically significant gender based differences
15	Colicino et al. (2017) [33]	USA	428 older men in the Veterans Affairs (VA) Normative Aging Study	Black carbon	MMSE Score Telomere length C Reactive Protein	Black Carbon associated with lower cognition. Each doubling in BC level associated with 1.57 (95% CI: 1.20, 2.05) times higher odds of low MMSE scores in individuals with longer blood Telomere length (OR = 3.23; 95% CI: 1.37, 7.59; *p* = 0.04 for BC-by-TL-interaction).
16	Forns et al. (2017) [34]	Barcelona, Spain	1439 of 2897 children recruited from 39 schools across Barcelona who participated in the BREATHE project 2012/2013	Composite exposure to indoor and outdoor levels of various TRAPs such as elemental carbon (EC), nitrogen dioxide (NO_2_), PM_2.5_ and ultrafine particles (UFP) at school	Working Memory	Slower development of working memory in children over 3.5 years period associated with higher schools based exposure to air pollution Did not report analysis stratified by sex
17	Lett et al. (2017) [35]	USA	Sub population of Early Childhood Longitudinal Study, Birth Cohort (*n* = 4050)	Isophorone exposure using 2002 National Air Toxics Assessment levels Home learning environment assessed with a modified Home Observation for Measurement of the Environment (HOME) Inventory	Standardized math assessment scores as a measure of early cognitive skills.	High isophorone levels (>0.49 ng/m^3^) and low HOME score were associated low math scale score. Decrement in math scale score was more than the additive effect of each exposure especially in male children.
18	Oudin et al. (2017) [36]	Umeå, Northern Sweden	1469 participants aged 60 to 85 years from Betula project followed up every five years from 1988 to 2010	Cumulative annual residential mean of NO_x_ (marker of long-term exposure to TRAP)	Episodic memory	No association between long-term exposure to air pollution especially TRAP and episodic memory. Did not report analysis stratified by sex
19	Stingone et al. (2017) [37]	USA	6900 children enrolled in the Early Childhood Longitudinal Study Birth Cohort	Residential concentrations of 104 ambient air toxins (including trichloroethylene, isophorone) from the National Air Toxics Assessment (2002) at age 9 months as per ZIP codes	Mathematics Tasks score	High isophorone levels (>0.49 ng/m^3^) were associated with low mathematics task scores in urban and highly populated urban areas Did not report analysis stratified by sex
20	Sunyer et al. (2017) [38]	Spain	2687 school children from 265 classrooms in 39 schools across Barcelona, Spain over one year from January 2012 to March 2013	Ambient TRAP exposure with daily levels of nitrogen dioxide (NO_2_) and elemental carbon (EC) in PM_2.5_ measured from PM_2.5_ filters at a fixed air quality background monitoring station and in schools.	Computerized child Attention Network test (ANT)	TRAP was associated with attentional impairment Daily ambient levels of NO_2_ and PM_2.5_ contributed to an estimated equivalent to a 1.1 month delay in age appropriate improvement in response speed as part of natural developmental trajectory. Boys had worse cognitive performance on Hit Reaction Time with incremental increase of both Elemental Carbon and NO_2_
21	Tallon et al. (2017) [39]	USA	3377 participants aged 57 to 85 years (from Wave 2, August 2010 to May 2011 in National Social Life, Health, and Aging Project (NSHAP) cohort study	PM_2.5_ exposure (estimated using GIS-based spatio-temporal models) and nitrogen dioxide (NO_2_) exposures (obtained from EPA monitors).	Chicago Cognitive Function Measure (CCFM)	High PM_2.5_ exposures associated with decrease in Chicago Cognitive Function Measure scores equivalent to aging by 1.6 years for PM_2.5_ and 1.9 years for NO_2_ exposure. Cognitive impact of PM_2.5_ was modified by stroke, anxiety, stress and mediated by depression.Cognitive impact of NO_2_ were mediated by stress with effect modification by impaired activities of daily living. Did not report analysis stratified by sex
22	Tzivian et al. (2017) [40]	Bochum, Essen, and Mülhei, Germany	Heinz Nixdorf Recall population based cohort study with 4086 participants	Land use regression was used to assess long-term residential concentrations for size-fractioned PM and nitrogen oxides. Assessment of road traffic noise	Cognitive assessment using five neuropsychological subtests and an additively calculated global cognitive score	Association of air pollutants with cognitive dysfunction was amplified by higher noise exposure at high levels of exposure. Did not report analysis stratified by sex
23	Best et al. (2016) [41]	USA	(*n* = 454; age ≥ 60 years from the 2001–2002 National Health and Nutrition Examination Survey).	Urinary 1-hydroxypyrene (indicator of PAH exposure)	Digit Symbol Substitution Test (DSST)	Dose-dependent 1% increase in urinary 1-hydroxypyrene associated with 1.8% poorer performance on DSST
24	Oudin et al. (2016) [42]	Sweden	1469 persons aged 60 to 85 years at inclusion in the Betula project and followed up to 22 years, five years apart between 1988 and 2010	Exposure to traffic-related air pollution	Dementia incidence	Participants in the group with the highest exposure to TRAP more likely to be diagnosed with dementia. Did not report analysis stratified by sex
25	Porta et al. (2016) [43]	Rome, Italy	474 children from a birth cohort of 719 newborns enrolled in 2003–2004 as part of the GASPII project evaluated at the age of 7 years	Air pollutants (NO_2_, PM_coarse_, PM_2.5_, PM_2.5_ absorbance) during pregnancy and at birth Maternal smoking	IQ assessed with Wechsler Intelligence Scale for Children-III	Traffic intensity in a 100 m buffer around home and an incremental 10 μg/m^3^ higher exposure of NO_2_ exposure in intra-uterine period was associated reduced verbal IQ and verbal comprehension IQ. Other pollutants also showed negative associations with much larger confidence intervals. Did not report analysis stratified by sex
26	Tzivian et al. (2016) [44]	Bochum, Essen, and Mülhei, Germany	Heinz Nixdorf Recall population based cohort study with 4086 participants aged 50–80 years old	Land use regression was used to assess long-term residential concentrations for size-fractioned PM and nitrogen oxides. Assessment of road traffic noise	Cognitive assessment using five neuropsychological subtests and an additively calculated global cognitive score (GCS)	Long-term exposures to AP and traffic noise are negatively correlated with four subtests including memory and executive functions and GCS in dose-dependent relationship independent of noise exposure e.g., an interquartile range rise in PM_2.5_ correlated with verbal fluency, labyrinth test, and immediate and delayed verbal recall. Did not report analysis stratified by sex
27	Tzivian et al. (2016) [45]	Vide supra	Vide supra	Vide supra	Diagnosis of Mild Cognitive Impairment based on five neuropsychological tests (Vide supra) and subjective memory complaint	Positive dose-dependent association between long-term PM_2.5_ exposure and mild cognitive impairment, mainly amnesic subtype (aMCI) Did not report analysis stratified by sex
28	Chen et al. (2015) [46]	USA	1403 community-dwelling older women aged 65–80 years without enrolled in the Women’s Health Initiative Memory Study (WHIMS), 1996–1998	Cumulative PM_2.5_ exposure in 1999–2006 Based on given residential histories and air monitoring data	MRI Brain	Greater PM_2.5_ exposures associated with significantly smaller white matter (WM) volumes in frontal and temporal lobes and corpus callosum (equivalent to 1–2 years of brain ageing), but not of gray matter or hippocampus
29	Harris et al. (2015) [47]	Eastern Massachusetts, USA	1109 mother-child pairs in Project Viva, a prospective birth cohort study in eastern Massachusetts (USA)	Prenatal and childhood exposure to TRAPs including black carbon (BC) and PM_2.5_ assessed by distance of residence from roadways and traffic density	Verbal and nonverbal intelligence, visual motor abilities, and visual memory assessed at mean age of 8 years	Children with a residence less than 50 m away from major highway had lower nonverbal IQ and lower verbal IQ and visual-motor abilities Equivocal findings for cross-sectional associations with major roadway proximity, prenatal and childhood exposure to traffic density and PM_2.5_, third-trimester and childhood BC exposures Did not report analysis stratified by sex
30	Peterson et al. (2015) [48]	New York, USA	40 children aged 7 to 9 years born to any of the 665 urban Latina (Dominican) or African American women women 18–35 years old who were not cigarette smokers or users of other tobacco products or illicit drugs, with initial prenatal care by the 20th week of pregnancy, and who were free of diabetes mellitus, hypertension, and known human immunodeficiency virus recruited between 1998 and 2006 through the local prenatal care clinics who had completed survey and who had a full range of prenatal PAH exposure levels; no or very low prenatal exposure to environmental tobacco smoke (classified by maternal report validated by cotinine levels of less than 15 µg/L in umbilical cord blood) and low chlorpyrifos exposure (below 4.39 pg/g) 20 children were randomly selected from above the median PAH level group and 20 from below the median PAH level group	Prenatal airborne PAH exposure by the sum of 8 Nonvolatile PAH: benzo[a]anthracene, chrysene/iso-chrysene, benzo[b]fluoranthene, benzo[k]fluoranthene, benzo[a]pyrene, indeno[1,2,3c,d]pyrene, dibenzo[a,h]anthracene, and benzo[g,h,i]perylene. measured with personal air monitoring of the mothers over a 48-h period in the third trimester of pregnancy but assumed to be index exposure for the entire gestation Prenatal PAH exposures validated against concurrent 2-week monitoring of residential air samples in the final trimester of pregnancy Postnatal PAH exposure by spot urine sample at age of 5 years for 9 metabolites (1-Hydroxynaphthalene, 2 Hydroxynaphthalene, 2-Hydroxyfluorene, 3-Hydroxyfluorene, 9-Hydroxyfluorene, 1 Hydroxyphenanthrene, 2-Hydroxyphenanthrene, 3-Hydroxyphenanthrene, and 4 Hydroxyphenanthren.)	CBCL WISC-IV MRI Brain	A dose-response inverse relationship between prenatal PAH exposure and reductions of the white matter surface in most of the frontal, superior temporal, and parietal lobes, as well as the entire rostrocaudal extent of the mesial surface, in the left but not the right hemisphere of the brain and reduced white matter in later childhood associated with slower information processing speed during intelligence testing and more severe externalizing behavioral problems such as ADHD and Conduct disorder. Postnatal PAH exposure correlated with white matter surface measures in bilateral dorsal prefrontal cortex bilaterally independent of prenatal PAH exposure Stronger inverse associations of prenatal PAH exposure with white matter surface measures in girls as compared to boys
31	Schiko-wski et al. (2015) [49]	Ruhr Area and Borken, Germany	4874 women from the SALIA cohort (aged 55 years at baseline) enrolled between 1985 and 1994 and followed up in 2006 (*n* = 2116) and 2008 (*n* = 834) of which complete information available (*n* = 789)	Particulate matter (PM) size fractions and nitrogen oxides (NO_x_) Traffic load	CERAD-Plus test Apolipoprotein E (ApoE) alleles	Air pollution was inversely related to visuospatial abilities on cognitive assessment with significant adverse association of traffic load in carriers of ApoE ɛ4 risk alleles.
32	Sunyer et al. (2015) [50]	Barcelona, Spain	2715 children aged 7 to 10 years from 39 schools in Barcelona	Chronic traffic air pollution [elemental carbon [(EC), nitrogen dioxide (NO_2)_, and ultrafine particle number (UFP; 10–700 nm)] measured twice during 1-wk campaigns both in the courtyard (outdoor) and inside the classroom (indoor)	*n*-back and the attentional network tests	Detrimental associations between Traffic related air pollution and cognitive performance were stronger in boys than in girls
33	Ailshire and Clarke (2014)	USA [51]	Cross sectional data of 780 non-Hispanic black and white men and women aged ≥ 55 years from the 2001/2002 Americans’ Changing Lives Study	PM_2.5_ using PA air monitoring data linked to respondents using census tract identifiers.	Tests of working memory and orientation	Exposure to high PM_2.5_ concentrations associated with 1.5 times greater error rate Male gender was associated with more cognitive errors
34	Ailshire and Crimmins (2014) [52]	USA	13,996 men and women aged 50 years or older from the 2004 HRS survey	Residence in areas with higher PM_2.5_ concentrations	Cognitive Function	Living in areas with higher PM_2.5_ concentrations was associated with worse cognitive function especially episodic memory No statistically significant gender based differences
35	Gatto et al. (2014) [53]	Los Angeles Basin, USA	1496 individuals (mean age of 60.5 years)	Air pollutants [O_3_, PM_2.5_ and nitrogen dioxide (NO_2_)]	Six domains of cognitive function and global cognition	Increased exposure to PM_2.5_ associated with lower verbal learning Exposure to NO_2_ > 20 ppb associated with lower logical memory Exposure to O_3_ above 49 ppb associated with lower executive function No air pollutant significantly associated with global cognition. No statistically significant gender based differences
36	Guxens et al. (2014) [54]	Europe (Netherlands, Germany, France, Italy, Greece, Spain)	Mother-, infant pairs recruited between 1997 to 2008 yielding a total sample of 9482 children from 6 European population-based birth cohorts—GENERATION R (The Netherlands), DUISBURG (Germany), EDEN (France), GASPII (Italy), RHEA (Greece), and INMA (Spain)	Prenatal exposure to air pollutants such as (NO_2_, NO_x_) in all regions and PM (PM_2.5_, PM_10_ and PM_coarse_) and PM_2.5_ absorbance in a subgroup using land-use regression models for the period 2008 to 2011 and then modelled for air pollutant profile for the exact pregnancy periods using background monitoring sites.	Assessment for cognitive and psychomotor development at 1 and 6 years of age.	Prenatal air pollution exposure during pregnancy, particularly NO_2_, was associated with dose-dependent reduction in psychomotor development but not cognitive development. Did not report analysis stratified by sex
37	Tonne et al. (2014) [55]	London, UK	2867 white men retired from work (mean age 66 years) from Whitehall II cohort	Particulate matter from traffic exhaust	Alice Heim 4-I test, 20-word free-recall test, semantic and phonemic verbal fluency	Higher PM_2.5_ of 1.1 μg/m^3^ was negatively associated with reasoning and memory but not verbal fluency and significant 5-year decline in standardized memory score.
38	Loop et al. (2013) [56]	USA	20,150 men and women enrolled in the REasons for Geographic And Racial Differences1 in Stroke (REGARDS) cohort	Satellite-derived estimate of PM_2.5_ concentration map	Cognition	No consistent increase in odds of cognitive impairment with every 10 µg/m^3^ increase in PM_2.5_ concentration. No statistically significant gender based differences
39	Power et al. (2013) [57]	USA	628 men (mean age of 70 years) from the VA Normative Aging Study	TRAP exposure on cognitive function.	HFE C282Y variant (hemochromatosis gene polymorphisms) Cognitive function	Older adults lacking HFE C282Y variant had greater adverse cognitive impact of TRAP exposure
40	Weuve et al. (2012) [58]	USA	19,409 US women aged 70 to 81 years from Nurses’ Health Study Cognitive Cohort	Long-term exposure to higher levels of both PM_2.5–10_ and PM_2.5_	Cognition	Dose-dependent association between Long-term exposure to higher levels of both PM_2.5–10_ and PM_2.5_ and significantly faster cognitive decline at 2 years. Every 10 μg/m^3^ increment in long-term (2 years) exposure to PM_2.5–10_ and PM_2.5_ cognitively equivalent to aging by approximately 2 years.
41	Power et al. (2011) [59]	USA	680 men (mean ± SD, 71 ± 7 years of age) from the U.S. Department of Veterans Affairs Normative Aging Study	Traffic-related air pollution including Black carbon (BC) exposure	Mini Mental Status Examination (MMSE)	Significant association between Black carbon (BC) exposure and lower MMSE score
42	Edwards et al. (2010) [60]	Krakow Poland	214 offspring of a cohort of pregnant, healthy, nonsmoking women of Krakow, Poland, between 2001 and 2006	Maternal 48-hr personal air monitoring to estimate foetal Polyaromatic Hydrocarbon air pollutant exposure and PAH estimation from maternal blood sample and/or a cord blood sample at the time of delivery	Raven’s Coloured Progressive Matrices (RCPM) at 5 years of age	Higher (above the median of 17.96 ng/m^3^) prenatal exposure to airborne PAHs associated with decreased scores on RCPM corresponding to an estimated average decrease of 3.8 IQ points. Did not report gender-based differences
43	Freire et al. (2010) [61]	Spain	210 boys from a population-based birth cohort from southern Spain	NO_2_	Cognitive development	No significant association between NO_2_ and cognitive development
44	Chen and Schwartz (2009) [62]	USA	1764 adult participants (aged 37.5 ± 10.9 years) of the Third National Health and Nutrition Examination Survey in 1988–1991	Geocoded Residential Ambient annual PM_10_ and ozone	Neurobehavioral Evaluation System-2 (NES2) data (including a simple reaction time test [SRTT] measuring motor response speed to a visual stimulus; a symbol-digit substitution test [SDST] for coding ability; and a serial-digit learning test [SDLT] for attention and short-term memory	Consistent associations between ozone and not PM_10_ and reduced performance on symbol-digit substitution test and a serial-digit learning test but not in simple reaction time test. Each 10-ppb increase in annual ozone was associated with equivalent to 3.5 and 5.3 years of aging-related decline in cognitive performance. Did not report analysis stratified by sex
45	Perera et al. (2009) [63]	New York, USA	249 Children born to nonsmoking black or Dominican-American women residing in New York City who had undergone prenatal ambient personal PAH monitoring	PAH exposure	Wechsler Preschool and Primary Scale of Intelligence-Revised (WPPSI-R)	High Prenatal Polyaromatic Hydrocarbons (PAH) levels (above the median of 2.26 ng/m^3^) inversely associated with full-scale IQ and verbal IQ with a decrement of 4.31 to 4.67 points of IQ at age of 5 years. Did not report analysis stratified by sex
46	Ranft et al. (2009) [64]	Germany	399 women aged 68–79 years	Traffic-related fine PM	Mild Cognitive Impairment (MCI)	Exposure to traffic-related fine PM consistent and significant risk factor for MCI
47	Lee et al. (2007) [65]	USA	278 children aged 12–15 years included in the National Health and Nutrition Examination Survey 1999–2000	POPs such as 3,3’,4,4’,5-pentachlorobiphenyl, 1,2,3,4,6,7,8-heptachlorodibenzo-p-dioxin (HPCDD), 1,2,3,4,6,7,8,9-octachlorodibenzo-p-dioxin (OCDD), 1,2,3,4,6,7,8-heptachlorodibenzofuran (HPCDF), beta-hexachlorocyclohexane, p,p’-dichlorodiphenyltrichloroethane, and trans-nonachlor.	Prevalence rates of learning disability (LD) and attention deficit hyperactivity disorder (ADHD), both of which are characterized by cognitive impairment	Direct association between POPs and LD/ADHD. No statistically significant gender based differences found
B	Stduies from Upper and Middle Income Settings					
1	Saenz et al. (2018) [66]	Mexico	13,023 Mexican adults over age 50 participating in 2012 Wave of the Mexican Health and Aging Study	Indoor air pollution (Use of wood or coal as primary cooking fuel)	Verbal learning, verbal recall, attention, orientation and verbal fluency	Exposure to indoor air pollution associated with poorer cognitive performance Did not report analysis stratified by sex
2	Zhang et al. (2018) [67]	China	25,486 individual respondents over age 10 in 2010 and 2014, from China Family Panel Studies (CFPS)	Cumulative and transitory exposures to air pollution	Verbal and Math tests	Adverse cognitive impact of air pollution on performance in verbal and math tests with greater deficits on verbal tasks as people aged. Significantly worse verbal tests performance in males on verbal tasks with short-term and cumulative air pollution exposure
3	Calderón-Garcidueñas et al. (2012) [68]	Mexico	30 children (20 from Southwest Mexico City (SWMC) and 10 from Polotitlan)	High pollution versus low pollution areas	Frontal tau hyperphosphorylation with pre-tangle material amyloid-beta diffuse plaques	Nearly 40% of highly exposed children and young adults had frontal tau hyperphosphorylation with pre-tangle material and 51% had amyloid-beta diffuse plaques compared versus 0% of controls living in low pollution areas. Did not report analysis stratified by sex
5	Calderón-Garcidueñas et al. (2011) [69]	Mexico	20 children from Mexico City (Mean age = 7.1 years, SD = 0.69) and 10 children from Polotitlán (Mean age = 6.8 years, SD = 0.66)	High pollution Areas versus Low pollution areas	MRI Brain Wechsler Intelligence Scale for Children-Revised (WISC-R) Inflammatory markers interleukin 12, cytokines and chemokines	Complex modulation of cytokines and chemokines influences children’s central nervous system structural and volumetric responses and cognitive correlates resulting from environmental pollution exposures Did not report analysis stratified by sex Exposure high pollution associated with low white matter volumes especially in the right parietal and bilateral temporal areas irrespective of white matter abnormalities as well as progressive deficits on WISC-R Vocabulary and Digit Span subtests
5	Calderón-Garcidueñas et al. (2011) [69] (Continued)					White matter hyperintensities associated with evidence of inflammation, immunoregulation, and tissue remodeling on MRI. Children without white matter hyperintensities had raised levels of interleukin 12, pro-inflammatory cytokines, and low levels of neuroprotective cytokines and chemokines. Did not report analysis stratified by sex
6	Calderón-Garcidueñas et al. (2008) [13]	Mexico	55 Children (mean age: 9.2 years) from Mexico City with high air pollution and 18 children (mean age: 10.5 years) from Polotitlán with low air pollution	Air Quality	Psychometric testing MRI Brain	Residence in high air pollution area associated with deficits in a combination of fluid and crystallized cognition tasks, high rates of prefrontal white matter hyperintense lesions Did not report analysis stratified by sex

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
