# Peer review of "Air Pollution and Cognitive Impairment across the Life Course in Humans: A Systematic Review with Specific Focus on Income Level of Study Area"

_ijerph, 2022, doi:10.3390/ijerph19031405_

Round 1

Reviewer 1 Report

The manuscript is a well-designed study concerning an important issue of cognitive impairment related to air pollution. However, some aspects might be improved.

In the introduction section in lines 62-64 there is information that „Cognitive disorders such as dementia disorders entail great suffering, high societal costs, and the prevalence worldwide is increasing. The number of people living with dementia is estimated to reach 75 million worldwide by 2030, with the majority living in low-income and middle-income countries.” Therefore, the information about number of people living with dementia in 2020 is missing. I suggest to include this information to compare the numbers and highlight the worldwide increase of this problem.

In the materials and methods section there is information that 84 papers are included in this systematic review, however the table include only 53 original research. I suggest to add the table with those 31 papers which are not original research or include in the study only those 53 original research.

There are also some spelling and editorial mistakes, which should be corrected. 

Author Response

Response to Reviewer 1 Comments and Suggestions for Authors

  1. The manuscript is a well-designed study concerning an important issue of cognitive impairment related to air pollution. However, some aspects might be improved.

In the introduction section in lines 62-64 there is information that „Cognitive disorders such as dementia disorders entail great suffering, high societal costs, and the prevalence worldwide is increasing. The number of people living with dementia is estimated to reach 75 million worldwide by 2030, with the majority living in low-income and middle-income countries.” Therefore, the information about number of people living with dementia in 2020 is missing. I suggest to include this information to compare the numbers and highlight the worldwide increase of this problem.

 Response: The suggestion has been accepted and the global prevalence of dementia (i.e. 55 million)has been incorporated in the manuscript as follows:

The number of people living with dementia is 55 million and is estimated to reach 75 million worldwide by 2030, with the majority living in low-income and middle-income countries. 

  1. In the materials and methods section there is information that 84 papers are included in this systematic review, however the table include only 53 original research. I suggest to add the table with those 31 papers which are not original research or include in the study only those 53 original research.

Response: The suggestion has been accepted. The manuscript includes only 53 original research papers in text. Figure 1 has also been revised to reflect this suggestion. The amended section is as follows:

A total of 1,173 studies with a keyword related to cognition and a keyword related to pollution (in their titles) were obtained till 10 July 2020. The remaining 350 studies were manually reviewed for relevance (Rai, Sandhu, Kumari) to yield  53 original research studies. (Figure 1)

  1. There are also some spelling and editorial mistakes, which should be corrected. 

Response: All spelling and editorial mistakes have been scrutinized and corrected.

Reviewer 2 Report

The manuscript consists of a systematic review of studies on exposure to environmental pollutants and neurocognitive effects according to income. The review is of good quality and demonstrates the relationship between exposure to environmental pollutants and cognitive impact in lower-income groups, for example, due to exposure to wood-burning fireplaces. The document provides data to make decisions that allow reducing exposure to these pollutants, especially in early childhood and the prenatal period.

Introduction:

- In the manuscript, the authors declare: we aimed to systematically review the evidence base concerning the relationship between air pollution and cognitive health outcomes, including dementia across the life-course. The review aims to focus on the country's income level of the study areas since LICs and LMICs often previously have been left behind when it comes to epidemiological studies of air pollution health effects. I suggest specifying the objective. It could be: "to systematically review the evidence base concerning the relationship between air pollution and cognitive health outcomes including dementia across the life-course according to the country's income level."

Materials and method:

- Authors must indicate the period that they are going to contemplate the publications for the review. The last ten years? No time restriction?

- PRISMA graph: Figure 1 must use the prism method to describe the identification of studies via databases and registers, reporting records removed before the screening, screening with records screened, records excluded, reports sought for retrieval, reports assessed for eligibility, reports excluded, and finally the studies included in review and descriptions of included studies. See the following address where the PRISMA flow diagram file is: http://www.prisma-statement.org/PRISMAStatement/FlowDiagram I also attach the file so that you can work in that format, figure 1

Results:

- The authors could describe the results by percentages according to exposure or outcome, or according to LICs or LMIC.

- Table 3 could go in an annex or supplementary file.

- Figure 2 and Figure 3 can only be described in the text and remove both graphs.

Discussion:

- I suggest adding in the discussion the quality of the evaluated manuscripts.

Author Response

  1. The manuscript consists of a systematic review of studies on exposure to environmental pollutants and neurocognitive effects according to income. The review is of good quality and demonstrates the relationship between exposure to environmental pollutants and cognitive impact in lower-income groups, for example, due to exposure to wood-burning fireplaces. The document provides data to make decisions that allow reducing exposure to these pollutants, especially in early childhood and the prenatal period.

Introduction:

- In the manuscript, the authors declare: we aimed to systematically review the evidence base concerning the relationship between air pollution and cognitive health outcomes, including dementia across the life-course. The review aims to focus on the country's income level of the study areas since LICs and LMICs often previously have been left behind when it comes to epidemiological studies of air pollution health effects. I suggest specifying the objective. It could be: "to systematically review the evidence base concerning the relationship between air pollution and cognitive health outcomes including dementia across the life-course according to the country's income level."

Response: The suggestion has been accepted. The objective has been incorporated in the manuscript as follows:

The aim and objective of this paper is to systematically review the evidence base with respect to the relationship between air pollution and cognitive health outcomes including dementia across the life-course and in diverse income settings. 

  1. Materials and method:

 Authors must indicate the period that they are going to contemplate the publications for the     review. The last ten years? No time restriction?

Response: As there is paucity of literature on the cognitive impact of air pollution, all publications on the topic with no time limit on date of publication were included in this systematic review. This was clearly stated in the Section 2.2 as

“ The inclusion criteria for the studies were full‐text articles published in English with no time limit on date of publication….” 

  1. - PRISMA graph: Figure 1 must use the prism method to describe the identification of studies via databases and registers, reporting records removed before the screening, screening with records screened, records excluded, reports sought for retrieval, reports assessed for eligibility, reports excluded, and finally the studies included in review and descriptions of included studies. See the following address where the PRISMA flow diagram file is: http://www.prisma-statement.org/PRISMAStatement/FlowDiagram I also attach the file so that you can work in that format, figure 1

Response: The suggestion is accepted and a new Figure 1 based on PRISMA flow chart format ( http://www.prisma-statement.org/PRISMAStatement/FlowDiagram ) is incorporated in the manuscript and the earlier Figure 1 deleted.

  1. Results:

- The authors could describe the results by percentages according to exposure or outcome, or according to LICs or LMIC.

Response: The original studies focussed on diverse exposure and outcome variables for differing population subsets as per age and income level of the site of the study. It was not possible to synthesize this heterogenous data from different studies and hence descriptive statistics like frequencies and percentages could not be calculated. This was clearly stated  in Section 2.4 on Data Extraction and Data Analysis

Due to heterogeneity in exposure and outcome variables and tools employed to assess these parameters, a statistical meta-analysis was not possible. Instead, the evidence was analyzed and presented based on  nature of air pollutant, age profile and gender of participants as well as income levels of study areas. 

The heterogenous nature of published data not amenable to pooling for statistical analysis  was also emphasized in the section on Quality Assessment.

Statistical methods for assessing publication bias using techniques like funnel plots were considered but ruled out due to heterogenous nature of available scientific literature on multiple exposures for multiple cognitive and neuroimaging outcomes across different age groups.

In light of this, the data was classified primarily as per age, followed by nature of exposure and outcome variables tested and the income level of the site of the study.

-5. Table 3 could go in an annex or supplementary file.

Response: The suggestion is not accepted as Table 3, though long, summarizes the diverse findings adverse cognitive impact on air pollution across life course. The suggestion of Reviewer No 3 on this issue to reformat the table so that headings are visible on all pages has been accepted.  

6, Figure 2 and Figure 3 can only be described in the text and remove both graphs.

Response: The suggestion for Figures 2 which has been placed as Annexures 2. However, Graph 3 helps visualize the lack of focus on gender when the researchers, who investigated the cognitive impact of air pollution, designed the methodology and analyzed their results. Hence, it is included in the main manuscript and re-labelled as Graph 2.

  1. Discussion:

- I suggest adding in the discussion the quality of the evaluated manuscripts.

 Response: The suggestion is accepted and the following paragraph has been incorporated in the discussion.

The quality of individual studies evaluated using Revised tool for Quality Assessment on Diagnostic Accuracy Studies (QUADAS -2) [21] revealed that most research  scored high on the quality of primary diagnostic accuracy. The process of QUADAS 2 evaluation, by involving different researchers than the ones involved in primary screening, ensured objectivity. The studies scored low on risk of bias and applicability concerns for patient selection, index test, reference standard, flow of patients through the study, timing of the index test(s) and reference standard. This indicates that the diverse research on cognitive impact of air pollution across life course and across different income levels of sites of study is of good quality and can inform policy decisions.

Reviewer 3 Report

Highlight changes in yellow in a next revision, please. No track changes.

The manuscript reveals significant similarity, in some cases with sentences that need to be from the author… describing what was done.

There cannot be references in abstract: “conducted, using PRISMA Guidelines [1],”

All abbreviations need to de defined first, abstract and text… as usual… “LMICs”

The English is not comprehensible…

“ No study from LMICs was identified despite high 29 levels of air pollutants and high rates of dementia. To conclude, air pollution may impair cognitive 30 function across the life course, but paucity of studies from LMICs is a major lacuna in research.”

I would like to see clear findings in a review advancing knowledge…

References cannot stat by number two…

1. Introduction 35

The Global Burden of Diseases, Injuries, and Risk Factors Study 2015 identified 36 ambient air pollution as a leading cause of the global disease burden, especially in 37 low-income and middle-income countries [2].”

Revise formatting: “preventable.[11]”

Do not use our or we… “In this review, our aim”

Language is unclear (incorrect..) and content difficult to access:

2. Materials and Methods 85

A systematic review was conducted to answer the research question on the impact 86 of air pollution on cognitive health across life course across diverse income settings. The 87 review that took place between January and October 2020 based on Preferred Reporting 88 Items for Systematic reviews and Meta-Analyses (PRISMA) [1] Statement using a defined 89 protocol which is unpublished (Appendix 1).”

Do not use abbreviations in captions:

Table 1. PICO Framework for Research Question.”

Unclear: “The search string included keywords related to cognition and air pollution.”

?!:

“by three reviewers 116 (C.B.R., N.K., V.K.S.),” and more

And varies:

“was consulted (M.C.).”

“(MC and KC).”

The references in Table 2 need to correspond to reference numbers (added) as OTHERS…

Figure..

And unclear fomat and content, depication information…

Graph 1 Publication Table Analysis of Publication bias metrics for published literature 154 on cognitive impact of air pollution using QUADAS 2.

Why italics?! Is it a title ad comment at the same time. No etxt before Figure?!

3. Results 156

3.1. A total of 1,173 studies with a keyword related to cognition and a keyword related to pollution 157 (in their titles) were obtained till 10 July 2020. The remaining 350 studies were manually reviewed 158 for relevance (Rai, Sandhu, Kumari) to yield 84 studies. (Figure 1)

Timeframe?

Figure 1. PubMed search results for studies on Air Pollution and Cognition across Life 160 Course.”

The sae:

3.2. Income levels of sites of original studies: The 84 included papers comprised of 53 original 162 studies of which 47 were conducted in HICs, 6 were conducted in UMICs. These are summarized 163 in Table 3. 164

Table 3. Original research on air pollution and cognition across the life course.”

Authors cannot extend the table through so many pages, there are ways to address it  and repeat the first row too…

Table 3. Original research on air pollution and cognition across the life course.”

?! Any reference then?

Graph 2. Sites of original research on air pollution and cognition as per World Bank income levels.”

Why use upper letter all over?!

No grapgh but figure…

Graph 3. Gender-based reporting in original studies on air pollution associated cognitive impair-175 ment.”

Check again..

3.5. Analysis based on life course and income level of study areas: 177

The results of the studies were analyzed based on the life course stage (children and 178 adolescents: prenatal and post natal exposures, adults and older adults) and on the in-179 come level of the study areas are presented below. 180

3.5.1. Studies related to air pollution and cognitive impairment in children and adoles-181 cents

3.5.1. A. Prenatal exposure studies 183

Prenatal exposure to different air pollutants has been analyzed for cognitive and 184 other developmental indicators from infancy through adolescence using birth cohorts. 185

3.5.1. A.i. Multiple air pollutants”

3.5.1 duplicated all over

Authors need to turn this review relevant, it became  succession of a list..

(…)

HUGE text..

Needs reformatting:

Figure 2. Proposed aetio-pathological mechanisms of cognitive impact from air pollution across the 530 life course.”

Well, this is not acceptable. A relevant review relies in the conclusions, not o a succession of content from publications

  1. Conclusions 564

Air pollution has been shown to have an adverse cognitive impact across the life 565 course, and is a preventable risk factor. The regulation of air pollution exposure, there-566 fore, has potential health benefits and resultant cost-saving by potentially improving 567 cognitive health and reducing the risk of dementia later in life. However, research on this 568 issue is non-existent in LIC and LMIC, leading to important knowledge gaps on the 569 shape of the dose-response curve, as these settings often experience higher levels and 570 diverse sources of air pollution. Further life course research, especially in LIC and LMIC, 571 is needed to establish aetio-pathogenic pathways, which would support existing scien-572 tific evidence and help motivate air pollution mitigation strategies needed for larger 573 policy change. “

Brief contextualization and methodology needed, as main findings and practical implications

79 references in a review, in which authors are included, is hardly enough.

Author Response

Response to Reviewer 3 Comments and Suggestions for Authors

  1. Highlight changes in yellow in a next revision, please. No track changes.

  Response: The suggestion is accepted

  1. The manuscript reveals significant similarity, in some cases with sentences that need to be from the author… describing what was done.

 There cannot be references in abstract: “conducted, using PRISMA Guidelines [1],”

  Response: The suggestion is accepted. The aforesaid reference has been removed.

  1. All abbreviations need to de defined first, abstract and text… as usual… “LMICs”

 Response: The suggestion is accepted. LMIC in abstract has been replaced with Low and Middle Income Countries.  The manuscript has been screened again to ensure all abbreviations are defined when they first appear.

  1. The English is not comprehensible…

“ No study from LMICs was identified despite high levels of air pollutants and high rates of dementia. To conclude, air pollution may impair cognitive function across the life course, but paucity of studies from LMICs is a major lacuna in research.”

I would like to see clear findings in a review advancing knowledge…

 Response: The statement   “No study from LMICs was identified despite high levels of air pollutants and high rates of dementia” is supported in the main manuscript in the introduction section itself.

There are two aspects of this statement: “high levels of air pollutants” and “high rates of dementia”  in Low and Middle Income Countries (LMICs).

The high rates of air pollutants in LMICs is illustrated by the following paragraphs which quote referenced documents.

PM2.5 from fossil fuels alone were in another study estimated to be a contributing cause to 10.2 million global excess deaths in 2012, with 62% of deaths in China (3.9 million) and India (2.5 million) [4].

For example, in India (a LMIC), pollution is worse than in China (a UMIC). 22 Indian cities are in the global list of the 30 most polluted cities. Apart from urban sources of air pollution, the burning of agricultural stubble in nearby rural areas also contributes to burden of air pollution in Indian cities. In India, all indicators of air pollution furthermore greatly exceed WHO standards [8]; and concentrations are increasing [9]. Furthermore, African PM emissions often originate from old diesel-powered vehicles, poor household waste management, and households burning biomass are the predominant contributors to outdoor air pollution [10]. In order to reduce uncertainties in the estimates for LIC and LMIC, epidemiological studies in these countries are thus needed [7].

The high rates of dementia in LMICs is illustrated by the following paragraphs which quote referenced documents

The number of people living with dementia is 55 million and is estimated to reach 75 million worldwide by 2030, with the majority living in low-income and middle-income countries. Recent studies have reported a decline in the prevalence of dementia in high-income countries suggesting that dementia may, at least partially, be preventable.

The conclusion in the abstract “To conclude, air pollution may impair cognitive function across the life course, but paucity of studies from LMICs is a major lacuna in research.” Is supported by results of the systematic review.

  1. References cannot stat by number two…

 “ 1. Introduction 

The Global Burden of Diseases, Injuries, and Risk Factors Study 2015 identified 36 ambient air pollution as a leading cause of the global disease burden, especially in low-income and middle-income countries [2].”

Response: This error in numbering of referencing is because the PRISMA reference in the abstract was numbered 1 leading to the first reference in manuscript being numbered as 2. The suggestion is accepted. The references have been corrected.

  1. Revise formatting: “preventable.[11]”

 Response: The suggestion is accepted and formatting is revised.

  1. Do not use our or we… “In this review, our aim”

 Response: The suggestion is accepted and the sentence has been revised to

The aim and objective of this paper was to systematically review the evidence base with respect to the relationship between air pollution and cognitive health outcomes including dementia across the life-course.

  1. Language is unclear (incorrect..) and content difficult to access:

2. Materials and Methods 85

A systematic review was conducted to answer the research question on the impact 86 of air pollution on cognitive health across life course across diverse income settings. The 87 review that took place between January and October 2020 based on Preferred Reporting 88 Items for Systematic reviews and Meta-Analyses (PRISMA) [1] Statement using a defined 89 protocol which is unpublished (Appendix 1).”

Response:  The suggestion is accepted and this paragraph has been revised as follows:

A systematic review was conducted to answer the research question on the impact of air pollution on cognitive health across life course across diverse income settings. The exercise was conducted  between January and October 2020 using a defined protocol (unpublished; Appendix 1). The process of systematic review was as per preferred reporting Items for systematic reviews and meta-Analyses (PRISMA) statement.

  1. Do not use abbreviations in captions:

Table 1. PICO Framework for Research Question.”

Response:  The suggestion has been accepted and heading of Table 1 modified.

Table 1. The Population, Investigated Exposure, Comparison, Outcome (PICO) Framework for Research Question.

  1. Unclear: “The search string included keywords related to cognition and air pollution.”

?!:

 Response: The systematic review focused on two domains of air pollution and cognition each of which has separate constructs.  To ensure that the database search did not miss out relevant articles the key words were expanded to include constructs within the broad domain of air pollution and cognition. Hence even if air pollution and cognition were not the key words, the relevant articles were identified based on the components and constructs

  1. “by three reviewers 116 (C.B.R., N.K., V.K.S.),” and more

 And varies:

“was consulted (M.C.).”

“(MC and KC).”

Response: There seems to be some confusion regarding the authors involved in the process of search strategy and the authors involved in the process of quality evaluation of shortlisted studies using QUADAS 2

The relevant sections have been amended in the manuscript for greater clarity as follows:.

2.3. Screening Strategy 

The search strategy comprised of a two stage process, In the first stage three reviewers (C.B.R., N.K., V.K.S.) independently screened identified studies for eligibility by screening the titles and abstracts for study inclusion and exclusion criteria. References and citations of included papers were also reviewed to include additional potential articles. Abstracts of conference proceedings were searched for any relevant papers and posters.

The second stage comprised of screening the full-texts of studies for eligibility by the three aforesaid reviewers independently screened the full-texts of studies. If the reviewers did not agree, then a more experienced reviewer was consulted (M.C.).

In Section 2.5. Quality Assessment

Two senior authors, not involved in the initial screening of studies for inclusion in the systematic review (vide supra), independently evaluated primary studies using QUADAS 2 (M.C., K.C.) and any difference in agreement was resolved by consensus with the third experienced author (M,K,), not involved in initial screening of papers for eligibility criteria or quality evaluation. This ensured objectivity in the systematic review process.

  1. The references in Table 2 need to correspond to reference numbers (added) as OTHERS…

            Figure..

Response: The suggestion is accepted. References are added to Table 3

And unclear fomat and content, depication information…

  1. Graph 1 Publication Table Analysis of Publication bias metrics for published literature on cognitive impact of air pollution using QUADAS 2.

Response: This graph has been revised and the caption corrected.

  1. Why italics?! Is it a title ad comment at the same time. No etxt before Figure?!

3. Results

3.1. A total of 1,173 studies with a keyword related to cognition and a keyword related to pollution 157 (in their titles) were obtained till 10 July 2020. The remaining 350 studies were manually reviewed 158 for relevance (Rai, Sandhu, Kumari) to yield 84 studies. (Figure 1)

 Response: The italics were erroneous. This has been formatted appropriately now. The section has been amended as per suggestion of Reviewer 1 as follows:

A total of 1,173 studies with a keyword related to cognition and a keyword related to pollution (in their titles) were obtained till 10 July 2020. The remaining 350 studies were manually reviewed for relevance (Rai, Sandhu, Kumari) to yield  53 original research studies. (Figure 1)

  1. Timeframe?

Response: The time frame is clearly stated in Methodology Section 2.1 Search Strategy as follows:

A systematic search of the PubMed database was performed using PRISMA Guidelines with no time limit on the date of publication.

  1. Figure 1. PubMed search results for studies on Air Pollution and Cognition across Life Course.”

Response: Figure 1 has been revised as per PRISMA format as advised by Reviewer 2.

 “3.2. Income levels of sites of original studies: The 84 included papers comprised of 53 original 162 studies of which 47 were conducted in HICs, 6 were conducted in UMICs. These are summarized 163 in Table 3. 164

Response: This section has been revised as per suggestion of Reviewer 1 as follows:

3.2. Income levels of sites of original studies: The 53 original studies included 47 conducted in HICs and 6 conducted in LMICs. These are summarized in Table 3.

  1. Table 3. Original research on air pollution and cognition across the life course.”

Authors cannot extend the table through so many pages, there are ways to address it  and repeat the first row too…

Table 3. Original research on air pollution and cognition across the life course.”

?! Any reference then?

Response: The table has been modified and formatted as per the suggestion given. The citations for each study have been included in the table

  1. Graph 2. Sites of original research on air pollution and cognition as per World Bank income levels.”

Why use upper letter all over?!

Response:  World Bank is a specific institution and hence both words have been capitalized

The graph has been shifted to Annexure 2 as per suggestion of Reviewer 2.

  1. No grapgh but figure…

Graph 3. Gender-based reporting in original studies on air pollution associated cognitive impairment.”

 Check again..

Response: The graph has been checked again. It has been shifted to Annexure 2 as per suggestion of Reviewer 2.

  1. 5. Analysis based on life course and income level of study areas: 177

The results of the studies were analyzed based on the life course stage (children and 178 adolescents: prenatal and post natal exposures, adults and older adults) and on the income level of the study areas are presented below.

3.5.1. Studies related to air pollution and cognitive impairment in children and adolescents

3.5.1. A. Prenatal exposure studies

Prenatal exposure to different air pollutants has been analyzed for cognitive and other developmental indicators from infancy through adolescence using birth cohorts.

3.5.1. A.i. Multiple air pollutants”

3.5.1 duplicated all over

Response: Manuscript checked again. There is no duplication of findings but similar air pollutants have been investigated for prenatal and post natal exposure.

  1. Authors need to turn this review relevant, it became  succession of a list..

(…)

 HUGE text..

Needs reformatting:

Figure 2. Proposed aetio-pathological mechanisms of cognitive impact from air pollution across the 530 life course.”

 Well, this is not acceptable. A relevant review relies in the conclusions, not o a succession of content from publications

 Response: The suggestion is accepted and the manuscript has been revised accordingly by summarizing and contextualizing  findings in each section.

  1. Conclusions 

 Air pollution has been shown to have an adverse cognitive impact across the life-course, and is a preventable risk factor. The regulation of air pollution exposure, therefore, has potential health benefits and resultant cost-saving by potentially improving cognitive health and reducing the risk of dementia later in life. However, research on this issue is non-existent in LIC and LMIC, leading to important knowledge gaps on the shape of the dose-response curve, as these settings often experience higher levels and diverse sources of air pollution. Further life course research, especially in LIC and LMIC, is needed to establish aetio-pathogenic pathways, which would support existing scientific evidence and help motivate air pollution mitigation strategies needed for larger policy change. “

Brief contextualization and methodology needed, as main findings and practical implications

 Response: The suggestion is accepted and the section is rewritten as follows”

This  systematic review of all published original studies on cognitive impact of air pollution across the life course( using PRISMA guidelines) demonstrates different air pollutants have an adverse cognitive impact across the life-course. Most research is confined to HICs and UMICs with no original research in LMICs and LICS despite high rates of both air pollutants and dementia in these settings. Hence the  extrapolations of the dose-response curve based on data from HICS and UMICs may not be applicable for LMICs and LICs. As air pollution is a preventable risk factor, its regulation has potential health benefits. Further life course research, especially in LIC and LMIC, is needed to establish aetio-pathogenic pathways, which would support existing scientific evidence and help motivate air pollution mitigation strategies needed for larger policy change.

  1. 79 references in a review, in which authors are included, is hardly enough.

Response: There is limited research on cognitive impact of air pollution across life course. As per protocol , all publications without time limit were included in the systematic review so that no relevant reference is left out. 3 additional references have been incorporated. The limited number of references reflect paucity of work in the topic of cognitive impact of air pollution across life course. The only author whose work is cited is the last author Anna Oudin, who has done seminal work in the field of air pollution.

Round 2

Reviewer 3 Report

Highlight changes in yellow in a next revision, please. No track changes.

The authors have not answered, at all, to my similarity comment

The references now contained in tables do not respect the reference format towards the “()” used, and add space between authors names and “[]”

No “Graph” but Figure...

And add legends to yy axis, as usual

A figure IS NOT an analysis…

I believe the authors made minor changes and that this review does not obey to the laws of a insightful review.

Thy mostly ignored the aim of previous comments.

Author Response

  1. Highlight changes in yellow in a next revision, please. No track changes.

Response: The changes were highlighted in yellow as per the reviewer’s initial comment in the first round of review. In fact, the editorial office was mailed on 25 November to clarify whether I should highlight changers in yellow as the portal recommended track changes document. As advised by the editorial office, I had submitted the  document wherein the changes were highlighted in yellow.  First revised document after first round of peer review is attached for reference.

The first round of revisions were highlighted in yellow and the second round of revisions are highlighted in blue.

  1. The authors have not answered, at all, to my similarity comment

Response: In the response to reviewer’s comments the similarity comment was addressed as follows

Manuscript checked again. There is no duplication of findings but similar air pollutants have been investigated for prenatal and post natal exposure.”

To elaborate further, there are a limited number of air pollutants which have been investigated for cognitive impact in different age groups across the life course like Particulate Matter, Traffic Related Air Pollutions, Isophorone, Persistent Organic Pollutants, Polyaromatic Hydrocarbons.

Hence, the role of similar pollutants is described clearly in the review paper with pertinent studies for each age group. There is no duplication as these are different studies in different settings wherein specific cognitive domains have been assessed and age-appropriate cognitive testing and neuroimaging markers have been used. 

Regarding the cognitive impact of air pollutants in children, two kinds of exposures have been studied

  1. Prenatal exposure i.e. exposure of air pollutants by the foetus in utero which is evaluated by taking samples from the pregnant mother. In this scenario the effect of air pollutants is on the brain while it is developing in utero. Such studies are difficult to conduct but provide robust evidence base.
  2. Post natal exposure i.e. exposure to the infant after birth and during childhood and adolescence. In this scenario the effect of air pollutants is on the brain as it grows and matures.

This classification clearly elucidates the adverse cognitive impact of different air pollutants which is often cumulative and may manifest till adolescence.

  1. The references now contained in tables do not respect the reference
    format towards the “()” used, and add space between authors names and “[]”

Response: The round brackets () have been replaced by Square Brackets [] and space has been added between names of authors and “[]”,

All references in the table, body of manuscript and final list of references at the end of the paper have been manually checked for errors.

  1. No “Graph” but Figure...

Response: The word graph has been changed to Figure

  1. And add legends to yy axis, as usual

Response: The axis legends have been added in Figure 2

Y axis: Number of Studies

X Axis: QUADAS 2 domains

  1. A figure IS NOT an analysis…

Response: The caption has been changed to :

Figure 2 Gender based reporting in original studies on air pollution associated cognitive impairment

The graph has been changed to pie diagram for better understanding.

  1. I believe the authors made minor changes and that this review does not
    obey to the laws of a insightful review.

Response: A large number of changes incorporating the suggestions of all the three reviewers were made and highlighted in yellow color for the first revision and blue color for the second revision

  1. They mostly ignored the aim of previous comments.

Response: All previous comments were duly addressed. At no point were the aim of any of the previous comments ignored in the first round of revision. Still, it is hoped that in the second round of revision, the comments made by all the reviewers are adequately honored.
